# BOOSTING RECOVERY IN TRANSFORMER-BASED SYMBOLIC REGRESSION

## ABSTRACT

The traditional objective in regression is generalization. That is, learning a function from training data that performs well beyond the training data. Symbolic regression adds another objective, namely, interpretability of the regressor. In the context of regression, interpretability means that the representation of the regressor facilitates insights into mechanisms that underlie the functional dependence. State-of-the-art symbolic regressors provide such insights. However, the state of the art predominantly incurs high costs at inference time. The recently proposed transformer-based end-to-end approach is orders of magnitude faster at inference time. It does not, however, achieve state-of-the-art performance in terms of interpretability, which is typically measured by the ability to recover ground truth formulas from samples. Here, we show that the recovery performance of the end-to-end approach can be boosted by carefully selecting the training data. We construct a synthetic dataset from first principles and demonstrate that the capacity to recover ground truth formulas scales with the available computational resources.

## 1 INTRODUCTION

Given labeled training data, regression is the task to estimate a functional dependency between independent variables, typically denoted as $x$, and dependent variables, typically denoted as $y$. The key objective in regression is, as in most of machine learning, *generalization*, that is, achieving small prediction errors beyond the training data.

Symbolic regression adds a second objective, namely *interpretability* of the underlying model. Inspecting and interpreting the symbolic expression of a functional dependence can facilitate fundamental insights into its underlying mechanisms. Therefore, symbolic regression has found applications in almost all areas of the natural sciences (Udrescu & Tegmark, 2021; Cornelio et al., 2023; Camps-Valls et al., 2023), in engineering (Wu & Zhang, 2023; Abdusalamov et al., 2023; Tsoi et al., 2024), and in medicine (Christensen et al., 2022; La Cava et al., 2023b; Zhang et al., 2024).

We discuss both objectives on the *bar magnets* example by Strogatz (2000) from the field of ordinary differential equations (ODEs). It is given by two bar magnets on a common pin joint in the plane, attracted by spatially opposing north and south poles. From four different starting orientations of the magnets, we observe rotation angles $x_0$ and $x_1$ of the north poles of the two bar magnets and the change $y$ of the first magnet's angle over time. Solving the regression problem, means using the observations to find a function $f$ that satisfies

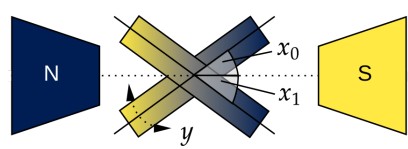

Figure 1: **Bar magnets example.**

$$y = \mathrm{d}x_0/\mathrm{d}t = f(x_0, x_1).$$

We use the observations to train a symbolic and a polynomial regressor. The generalization abilities of both regressors are illustrated in Figure 2. While both models perform well when predicting on unseen data, the symbolic model

$$y = -\sin(x_0) + 0.3 \sin(x_0 - x_1)$$

can be interpreted: The term $\sin(x_0 - x_1)$ can be interpreted as the value of the torque that drives the north poles of the two bar magnets apart. It is counteracted by the term $-\sin(x_0)$, that models

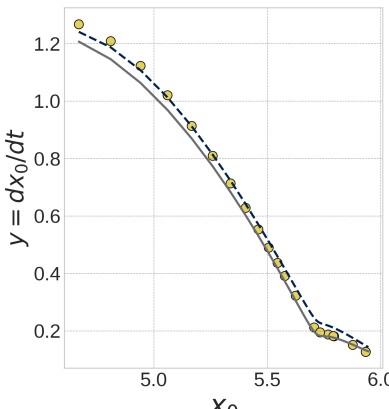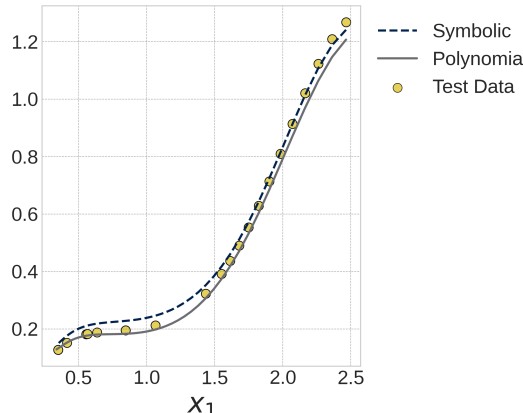

Figure 2: **Comparing the generalization ability of a polynomial and a symbolic regressor.** Both regressors perform well on *test* data, however, in different regimes of $x_0$ and $x_1$.

the value of the torque induced by the outer magnet. The polynomial model

$$y = -\,0.00903\,x_0^3 - 0.162\,x_0^2 x_1 - 0.042\,x_0^2 + 0.09\,x_0 x_1^2$$
$$+\,1.26\,x_0 x_1 + 0.917\,x_0 - 0.029\,x_1^3 - 0.258\,x_1^2 - 2.36\,x_1 - 1.77$$

is simply a linear combination of polynomials, which does not allow such an interpretation.

Given a problem instance, state-of-the-art symbolic regression methods either implicitly (Landajuela et al., 2021) or explicitly (Kahlmeyer et al., 2024) search a space of small expressions that are built from a predefined set of operations, usually represented by expression trees. Notably, the search is done at inference time and starts anew for each problem instance. Although symbolic regression aims for small expressions, because large expressions lose interpretability, search costs can be substantial, mostly because the space of expression trees grows exponentially with the number of variable and operator nodes in the expression trees (Virgolin & Pissis, 2022).

Biggio et al. (2021) proposed scalable neural symbolic regression with transformers to address the problem of compute-intensive inference by shifting the heavy lifting into a training phase. Indeed, inference with the resulting symbolic regressor can be orders of magnitude faster than state-of-the-art symbolic regressors. Kamienny et al. (2022) extends this work from three to up to ten input variables. Their approach performs well in terms of the standard regression measures model fit and model complexity. It does, however, perform poorly in terms of the standard measure of interpretability, namely, the ability to recover formulas, up to symbolic equivalence, from sampled data. On the established symbolic regression benchmark SRBench (La Cava et al., 2021), the transformer approach as presented by Kamienny et al. (2022) recovers only a maximum of $1.59\%$ of the ground truth formulas, whereas state-of-the-art symbolic regressors can recover up to $55\%$ percent. Shojaee et al. (2023) trade off learning against search-based approaches by using a transformer to guide the search for symbolic expressions, which improves recovery at the cost of increased inference time. However, while being much slower at inference time, it still does not achieve state-of-the-art recovery. Follow-up work on pure transformer approaches by Lalande et al. (2023) and Vastl et al. (2024) again performs well in terms of standard regression performance measures, but poorly in terms of recovery. Given that even better accuracy can be achieved by conventional methods such as polynomial regression or neural networks, which are not designed to be interpretable, the question of whether transformers can also recover interpretable expressions is still open.

Here, we show that the subpar recovery rate achieved by the end-to-end symbolic regressor of Kamienny et al. (2022) is not a problem of the overall architecture, but of the training data. By systematically designing a synthetic training dataset that covers a diverse set of functions, we demonstrate that the recovery performance of the end-to-end approach is limited only by the available computational budget, while preserving the advantage of fast inference.

## 2 TRAINING DATA GENERATION

Data in a standard regression task is a set of labeled data points that is split into training and test data, or sometimes into training, validation, and test data. Standard regressors are typically evaluated by assessing their generalization ability, which is measured in terms of their prediction accuracy on the test data. Generalization is also important for symbolic regressors, but another important objective is interpretability. Interpretability can be measured in terms of model complexity, for instance, by the number of nodes in an expression tree. A more direct measure of interpretability is the ability of a symbolic regressor to recover a ground truth formula. However, just having labeled data is not enough to measure recovery. The function from which the labeled data have been sampled must also be explicitly known. Therefore, data for symbolic regression tasks are *explicitly given functions* together with *labeled data* that have been sampled from the functions. Moreover, the transformer-based end-to-end approach to symbolic regression does not only need the functions in the evaluation phase but also in the training phase.

Generating data for training and evaluating symbolic regressors thus comprises two tasks, namely, selecting a set of explicitly given functions and computing representative samplings from these functions. In the following, we describe how we address the two tasks.

### 2.1 REGISTER MACHINE PROGRAMS

The most commonly used representation of formulas in symbolic regression are expression trees (Lample & Charton, 2019; Kommenda et al., 2020; Petersen et al., 2021; Virgolin et al., 2021). In their end-to-end approach, Kamienny et al. (2022) also randomly sample expression trees. Expression trees, however, are not a particularly compact representations, mostly because common sub-expressions are not factored out. A more succinct representation that effectively handles common sub-expressions are expression DAGs (Kahlmeyer et al., 2024). Expression DAGs thus have the advantage of being expressive while remaining short. Here, we represent symbolic expressions by *register machine programs (RMPs)* that are structurally equivalent to expression DAGs. In comparison to expression DAGs, RMPs have the advantage that they are naturally represented in sequential form, which facilitates their straightforward integration into the transformer architecture.

**Register Machine Programs (RMPs).** Given a set of unary and binary instructions $I$ (see the supplement for details), input variables $x_0, \ldots, x_{D-1}$ and registers $a_0, \ldots, a_{L-1}$. A *register machine program* with input $x \in \mathbb{R}^D$, output $y \in \mathbb{R}$, and $L$ lines is a sequence of the form

$$y = \text{RMP}(x) = \left[ a_i := \text{inst}(A_i) \right]_{i=0}^{L-1},$$

where $\text{inst} \in I$ and $A_i \subseteq \{x_0, \ldots, x_{D-1}\} \cup \{a_0, \ldots, a_{L-1}\}$. At each line of the RMP, an intermediate result, designated $a_i$, is generated, where $i$ is the line number. The final intermediate result is then treated as the output of the RMP. As an example, consider the symbolic expression $x_0^2 + x_0 x_1$, which can be written as the following RMP with dimension $D = 2$ and $L = 3$ lines,

$$a_0 := \text{sq}(\{x_0\}), \quad a_1 := \text{mult}(\{x_0, x_1\}), \quad a_2 := \text{add}(\{a_0, a_1\}).$$

In the example, we have $A_0 = \{x_0\}$, $A_1 = \{x_0, x_1\}$, and $A_2 = \{a_0, a_1\}$.

Since many formulas in the application domains of symbolic regression include *constants*, we also allow constants in our RMPs. In principle, it is enough to include only one constant from which additional constants can be computed.

For training a transformer-based symbolic regressor, we should not use all RMPs, because there are many redundant RMPs that compute the same function $f : \mathbb{R}^D \to \mathbb{R}$. RMPs that compute the same function $f$ are called *equivalent*.

**Equivalence classes of RMPs.** Let $f : \mathbb{R}^D \to \mathbb{R}$ be a function. The equivalence class of $f$ is the following set of RMPs,

$$\left\{ \text{RMP} \mid \forall x \in \mathbb{R}^D, \ \text{RMP}(x) = f(x) \right\}$$

For instance, the RMPs

$$[a_0 := \text{add}(x_0, x_0), \ a_1 := \text{add}(x_0, a_0)] \quad \text{and} \quad [a_0 := \text{mult}(x_0, 2), \ a_1 := \text{add}(x_0, a_0)]$$

both belong to the equivalence class of the function $f(x_0) = 3x_0$.

Of course, it is enough to consider only one RMP from an equivalence class. Conceptually and practically, it makes sense to choose an element of minimal length from each equivalence class.

**Minimal RMPs.** An RMP is called *minimal* for its equivalence class if it has minimal length $L$ among all the RMPs in the class. Still, minimal RMPs need not be unique. If, for a given equivalence class, several minimal RMPs exist, then we call these programs *minimal alternatives*.

Our basic approach thus becomes to generate minimal RMP alternatives up to a given length, such that every equivalence class is covered at most once and thus, for a given computational budget, the number of covered functions is maximized.

## 2.2 Sampling and Standardization

For most RMPs, the corresponding functions $f : \mathbb{R}^D \to \mathbb{R}$ do not have a "representative" data sample. Therefore, we are facing the problem to sample the data samples themselves. Here, we adapt the sampling approach by Kamienny et al. (2022), who already recognized the need to maximize the diversity of data samples. A sample with $N$ data points in $D$ dimensions is generated as follows:

1. Sample a number of clusters $k \sim \mathcal{U}(\{1, \ldots, k_{\max}\})$ and $k$ weights $w_i \sim \mathcal{U}([0, 1])$, and normalize the weights so that $\sum_i w_i = 1$.

2. For each cluster $i \in N_k$, sample a centroid $\mu_i \sim \mathcal{N}(0, 1)^D$, a vector of variances $\sigma_i \sim \mathcal{U}([0, 1]^D)$, and a distribution shape $\mathcal{D}_i \in \{\mathcal{N}, \mathcal{U}\}$ (Gaussian or uniform).

3. For each cluster $i \in \{1, \ldots, k\}$, sample $\lfloor w_i \cdot N \rceil$ input points from $\mathcal{D}_i(\mu_i, \sigma_i)$, apply a random rotation sampled from a Haar distribution, and scale the input points such that their axis-aligned bounding box becomes $[-a, a]^D$, for a given scaling parameter $a > 0$.

4. Use the RMP to compute the function values $y_i = f(x_i)$ at all sample points $x_i, i \in N$.

5. Standardize the set of all sample points, that is, the inputs $x_i$ and outputs $y_i$, by subtracting their mean and dividing by the standard deviation along each dimension.

Our sampling approach differs from Kamienny et al. (2022) in the last step. Kamienny et al. (2022) only standardize the input points $\{x_i\}_{i=1}^N$ but not the output points $\{y_i\}_{i=1}^N$. Their model predicts an intermediate function $\hat{f}$, which is subsequently mapped back to the target domain via the inverse of the standardization process. However, as demonstrated in previous work, end-to-end symbolic regression approaches are prone to domain overfitting (d'Ascoli et al., 2022; 2023). That is, the performance of these models often drops significantly when evaluated on data outside the training domain. To mitigate this issue, we standardize both the inputs and the outputs by subtracting their mean and dividing by the standard deviation for each feature and for the output.

As a consequence of standardization, we cannot distinguish samples from RMPs that differ by input and output translations and scalings. Therefore, we call two RMPs affinely equivalent when they differ only by such translations and scaling.

**Affinely equivalent RMPs.** Two RMPs are called *affinely equivalent* if their outputs are identical up to *scaling* and *translation* transformations. Specifically, two register machine programs $\text{RMP}_1$ and $\text{RMP}_2$ are called equivalent if there exist constants $c_1, c_2, c_3 \in \mathbb{R}$ and $c_4 \in \mathbb{R}^D$ such that, for any input $x \in \mathbb{R}^D$, the output of $\text{RMP}_1$ can be transformed into the output of $\text{RMP}_2$ as follows

$$\text{RMP}_2(x) = c_1 \cdot \text{RMP}_1(c_3 \cdot x + c_4) + c_2.$$

That is, $c_1$ represents an output scaling factor, $c_2$ represents an output translation, $c_3$ represents an input scaling factor, and $c_4$ represents an input translation vector.

## 2.3 Register Machine Program Selection

In practice, we face computational limitations that hinder finding minimal RMPs. For instance, our equivalence definitions cannot be tested in practice, because an infinite number of points has to be checked. For that reason, we weaken the definitions and only require functional equivalence on a fixed finite number of sample points that are sampled uniformly at random from the interval $[-a, a]$, where $a > 0$ is the same scaling factor as before. Another practical limitation is that the number of minimal RMPs grows so fast in the input dimension and the program length that restricting ourselves

Figure 3: **RMP sampling procedure.** The sampling procedure has three steps: enumeration of minimal RMPs under the computational constraints, generating additional candidate RMPs by recombining existing RMPs, and verifying the succinctness of the candidates by various tests.

to minimal RMPs that can be enumerated in practice excludes many practically interesting functions from the training set.

There are, of course, infinitely many RMPs. Therefore, we have to restrict ourselves to RMPs up to some maximal length. However, the number of RMPs of length at most $L$ still grows exponentially in $L$. Specifically, for $D$-dimensional inputs, the number of RMPs with $L$ lines satisfies the recursion

$$S_D(1) = D\,|I_1| + \binom{D}{2}|I_2|$$

$$S_D(L) = S(L-1)\left((D+L-1)|I_1| + \binom{D+L-1}{2}|I_2|\right),$$

where $|I_1|$ and $|I_2|$ are the numbers of unary and binary instructions in the instruction set $I$. If we unroll this recursion, we get the following closed-form expression,

$$S_D(L) = \prod_{j=0}^{L-1}(D+j)\left(|I_1| + |I_2|\frac{D+j-1}{2}\right),$$

which grows exponentially in $L$. Moreover, any RMP with input dimension $D$ has length at least $D-1$.

Remember that we want to build our training set from minimal RMPs. To make sure that an RMP is minimal, we need to know all RMPs with the same input dimension but smaller length. Assume that, in practice, we can look at only $10^9$ RMPs. Then, for input dimension $D=1$, we can exhaustively enumerate RMPs up to length $L=7$, and for input dimension $D=6$ RMPs up to length $L=5$. Enumerating RMPs with input dimension $D>6$ is not feasible. See the supplementary material for more details.

For our training dataset, we enumerate all minimal RMPs that can be found by enumerating $10^9$ RMPs for each dimension $D \in \{1,\ldots,6\}$. However, that does not cover all RMPs describing interesting phenomena in physics, such as, for instance, Washburn's formula (Washburn, 1921),

$$L(\gamma, D, t, \theta, \eta) = \sqrt{\frac{\gamma \cdot D \cdot t \cdot \cos(\theta)}{2 \cdot \eta}},$$

which has input dimension $D=5$ and length $L=7$. It is not included in our training set, because for input dimension $D=5$, we can only exhaustively enumerate RMPs up to length $L=5$ if we enumerate at most $10^9$ RMPs for any input dimension. Therefore, we add another sampling procedure that non-exhaustively samples succinct, but not necessarily minimal, RMPs of length $L>5$.

The extended sampling procedure is illustrated in Figure 3. It works as follows: Sample two minimal RMPs uniformly at random from the set of minimal RMPs of length up to five that we have

exhaustively enumerated before. Assume that the second RMP has input dimension $D$. Connect the output of the first RMP to a randomly selected input variable of the second RMP. Then sample, again uniformly at random, $D-1$ variables from the set of input and intermediate variables of the first RMP, and connect them to the remaining $D-1$ input variables of the second RMP. The result is again an RMP. We discard this RMP if its length exceeds a maximum value $L_{\max}$. Otherwise, for every non-output variable of the new RMP, we make sure that its derivative with respect to every input variable does not vanish. A vanishing gradient indicates that the instruction line of the RMP corresponding to the non-output variable eliminates input variables. As for instance in $x/x$ or $\sin^2(x) + \cos^2(x)$, which both evaluate to the constant 1. Similarly, for the output variable, make sure that the derivatives with respect to all intermediate variables do not vanish, because otherwise, we know that an intermediate variable does not contribute to the program and therefore is redundant, which means that an equivalent shorter RMP exists. Finally, use the equivalence test to ensure that the function represented by the newly sampled RMP is not covered by another RMP in the sample set. If it is already covered, keep the smaller RMP. We sample new RMPs until a given maximum number of equivalence classes is covered.

### 2.4 Sampling of Training Data

For training a transformer-based symbolic regressor with stochastic gradient descent, we need to sample batches of RMPs and then input-output examples from the sampled RMPs. Since the number of RMPs grows exponentially in $L$, sampling from all RMPs uniform at random would be biased towards large RMPs. For an unbiased sampling, generated RMPs are placed in $(D, L)$ buckets, where $D$ ranges from 1 to 10 and $L$ from 1 to 12. Remember, that buckets with $L < D - 1$ cannot hold an RMP. Feasible $(D, L)$ buckets contain minimal alternative RMPs from different equivalence classes. Now, we can sample batches of RMPs by first sampling a feasible $(D, L)$ bucket uniformly at random and then an RMP from the bucket. In the sampled RMPs, constant placeholders are replaced with constants drawn uniformly at random from a finite interval $(-b, b)$. For each sampled RMP, input-output examples are then sampled as described in Section 2.2.

## 3 Experiments

The goal of the experiments is examining the impact of our training dataset on the *recovery* performance of transformer-based symbolic regression. First, in Section 3.1 we explain our experimental setup. Then, in Section 3.2, we benchmark the recovery of a model trained on our training dataset against the state-of-the-art. Finally, in Section 3.3, we perform ablation studies to assess how the model components affect the recovery.

### 3.1 Experimental Setup

All experiments were conducted on a single NVIDIA RTX A6000 GPU.

**Tokenization, embedding, and architecture.** The encoder-decoder transformer architecture employed in our experiments follows the one described in Kamienny et al. (2022). For the encoding of the data points $(x, y) \in \mathbb{R}^D \times \mathbb{R}$, numbers are represented in base 10 floating-point notation, rounded to four significant digits, and encoded as sequences of three tokens Charton (2021). These tokens are the sign, the mantissa (between 0 and 9999), and the exponent (from `E-100` to `E100`). For example, the number 0.3 is encoded as `[+, 3, E-1]`. Therefore, each data point $(x_i, y_i)$ is represented by $3(D + 1)$ tokens.

RMPs are written sequentially line by line from the first instruction to the last instruction. Instructions from the instruction set and intermediate results $a_i$ are each represented by a single token. Constants are represented by placeholders without any indication of their numerical value. For example, the RMP for the function $f(x_0, x_1) = -\sin(x_0) + 0.3\sin(x_0 - x_1)$ is encoded as:

```
[neg, x_1, =, a_0, add, x_0, a_0, =, a_1, sin, a_1, =, a_2,
 mult, c_0, a_2, =, a_3, sin, x_0, =, a_4, neg, a_4, =, a_5,
 add, a_3, a_5, =, a_6]
```

Since each RMP line has at most five tokens, and we only consider RMPs of length at most twelve, we need 60 tokens for the RMP, or 62 tokens when we also include start and end tokens.

As in Kamienny et al. (2022), the tokens for each data point $(x_i, y_i)$ are first concatenated and then fed into a two-layer perceptron, which projects each data point down into the embedding dimension $d_{emb}$, with $d_{emb} = 512$ in our experiments. The resulting $N$ embeddings of dimension $d_{emb}$ are then fed into a standard transformer encoder stack with four layers. Given that input points for a multivariate regression problem do not naturally adhere to sequential order, we do not use positional embeddings within the encoder.

The RMP tokens are embedded into $d_{emb}$-dimensional space using a standard embedding layer. The embeddings are then fed into a standard transformer decoder stack. During the experiments, the decoders are varied from one to 16 layers. Moreover, experiments were conducted with input lengths of 192, 448, and 960 data points, respectively. The largest model has 86M parameters.

**Training.** The overall training strategy follows Kamienny et al. (2022). Here, we only highlight the differences, but a summary of all important training parameters can be found in the supplement. To avoid bias in the validation dataset, we withhold four equivalence classes for each dimension-length combination $(D, L)$ with $L > 1$. This provides us with a *validation* and a *test* dataset with 82 equivalence classes and 164 RMPs each. Models are trained until the cross-entropy loss on the validation set is saturated.

**Inference.** In contrast to Kamienny et al. (2023), we have to standardize the input and output dimensions of the data points, before they are tokenized and fed into the transformer. Then, $k$ candidate RMPs are generated by a beam search with beam size $k$. If a generated RMP contains a constant placeholder, then the placeholder is replaced by a constant that is fitted with the adapted Broyden–Fletcher–Goldfarb–Shanno (BFGS) algorithm by Nawi et al. (2006). To correct for the standardization, scaling and translation constants are also fitted by the adapted BFGS algorithm. After, fitting the constants, the $k$ candidate RMPs are ranked based on the $R^2$-score (Pearson, 1909). The highest scoring RMP is returned as the result of the inference.

## 3.2 SRBench Results

We benchmark the transformer-based symbolic regressor that is trained on our dataset against the state-of-the-art symbolic regression methods from the SRBench test suite (La Cava et al., 2021). SRBench supports the evaluation of symbolic regressors on a set of 252 regression problems and provides results for 14 state-of-the-art regressors. The regression problems themselves are divided into two groups: 130 ground truth problems, where the true underlying formula is known, and 122 black box problems, where only the samples are given.

Here, we focus on the recovery of ground truth formulas, because we regard recovery as the most direct measure of interpretability for symbolic regressors. We take the 130 ground truth formulas from the *Feynman Symbolic Regression Database* (Udrescu & Tegmark, 2020b) and the *ODE-Strogatz Repository* (La Cava et al., 2023a). Following the practice of La Cava et al. (2021), a ground truth formula $f$ is considered *recovered* by a regressor $\hat{f}$ if either $f - \hat{f}$ can be resolved symbolically to a constant, or $\hat{f}$ is non-zero and $f/\hat{f}$ can be resolved symbolically to a constant. We also report the *complexity* of expressions, which is measured by the number of nodes in corresponding expression trees. All symbolic checks are delegated to the Python library `SymPy` (Meurer et al., 2017).

**Recovery performance.** Figure 4 shows the recovery performance and complexity for the regressors included in the SRBench and the transformer approach trained with and without our training dataset. By using our training dataset while keeping the architecture fixed, the recovery performance of the transformer approach improves from $1.59\%$ (E2E) to $34.02\%$ (E2E-RMP). It now ranks third only behind the state-of-the-art search-based approaches UDFS (Kahlmeyer et al., 2024) and AIFeynman (Udrescu & Tegmark, 2020a), which require inference times that are orders of magnitude larger. More specifically, as can be seen in Figure 5, the average inference time per formula of the Feynman dataset is $0.32$ seconds, which is three orders of magnitude faster than state-of-the-art search-based approaches. The training data also impact the complexity of the formulas generated by the transformer approach. When not using our training data, the transformer approach generates complex formulas with multiple constants, whereas it generates formulas of lowest complexity among all regressors when using our dataset. Moreover, our training data makes the transformer approach more robust with respect to noise.

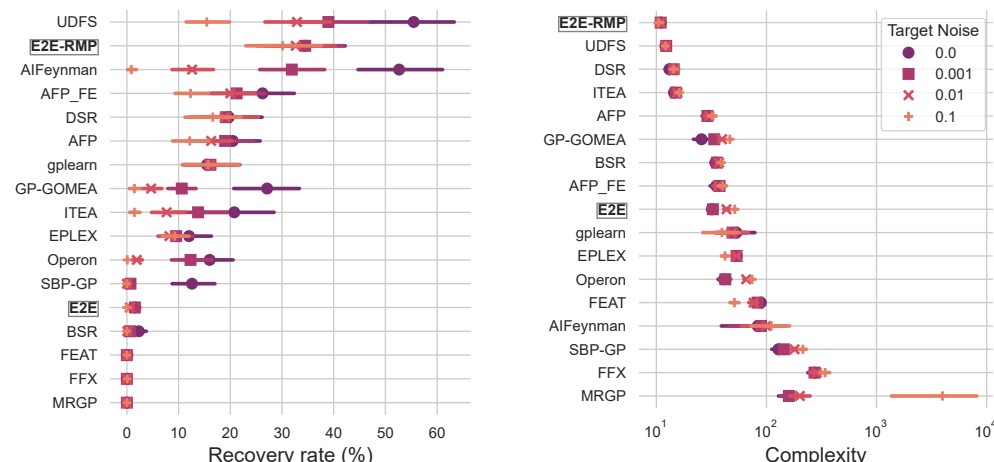

Figure 4: **SRBench results.** *Left:* Recovery rate of different state-of-the-art approaches for four noise levels. *Right:* Complexity of the predicted expressions as evaluated by the SRBench test suite.

## 3.3 ABLATION STUDIES

We perform ablation studies to assess the influence of the parameter size, the size of the training dataset, the beam size, and the input length on recovery performance. We compare the results on three datasets: 164 test RMPs sampled uniformly at random from all feasible $(D, L)$ buckets *not* used for training, 164 *training* RMPs sampled uniformly at random from the feasible buckets, and the 119 formulas from the *Feynman Symbolic Regression Database*. Results are shown in Figure 5.

**Number of parameters.** The recovery performance on the Feynman dataset shows a linear increase from the low $40\%$ to $50\%$ as the number of parameters increases. For attributing the recovery performinance to the *memorization* and the *generalization* capabilities of the model, we observe that $51\%$ of the expressions of the Feynman dataset are in the training dataset. That is, successful recovery becomes a mixture of memorization and generalization. Furthermore, $82.6\%$ of the recovered formulas are in the training dataset and could thus be considered successfully memorized. The remaining $17.4\%$ of the recovered formulas are not among the training data and are therefore the result of generalization. Memorization in our context is significantly more complex than mere memorization of a fixed sample of inputs and corresponding outputs, because, for a given function, input-output pairs are sampled with a highly diverse sampling strategy. This makes it improbable that the model has encountered the test instance during training. Moreover, linearly increasing recovery is also observed on both the test and training sets. The recovery scores, however, are significantly lower than the score on the Feynman dataset. This likely is because the test and training RMPs have been sampled uniformly from all feasible $(D, L)$ buckets. We show in the supplement that the difficulty of test instances increases with $D$ and $L$. Therefore, there are more difficult test instances among the test and training RMPs than among the Feynman formulas, which are typically smaller.

**Size of training dataset.** As can be seen in Figure 5, recovery increases linearly with the number of RMPs that ranges from one million to 16 million. That is, the number of RMPs and thus equivalence classes of functions seen during training has a significant impact on recovery. For a better understanding, we compare our data generation method to a generation method that samples RMPs as random instruction sequences, and to the expression tree generation method by Kamienny et al. (2022). For a comparison, we count the number of equivalence classes covered by the respective data generation methods out of 10 000 sampled RMPs. The random instruction sequence method covers only 760 different equivalence classes, and the expression tree sampling method by Kamienny et al. (2022) covers about 6 400 equivalence classes.

**Beam size and input length.** As can be seen also in Figure 5, recovery improves significantly when more than one beam is used. It saturates around 16 beams. Moreover, recovery degrades with increasing input length.

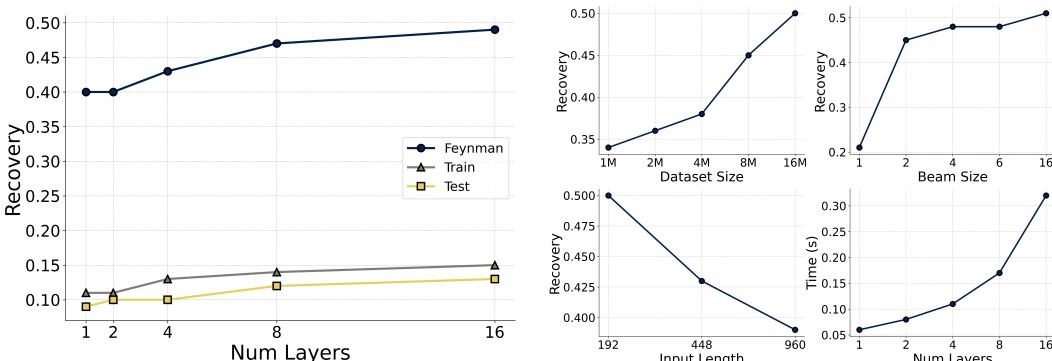

Figure 5: **Ablation studies.** *Left:* Comparison of the recovery performance on the Feynman, training and test datasets. *Right:* Comparing the effects of dataset size, beam size, and input length on recovery as well as inference times for a 16-layer transformer model.

## 3.4 Findings

From our experimental results, we derive the following three findings that facilitate the training of transformer-based symbolic regressors that perform well in terms of recovery.

**Training data matter.** Our experimental results show that the careful selection of synthetic data is key to successful end-to-end learning. This supports similar findings in works on large language models (LLMs), such as, for instance, in (Abdin et al., 2024). Moreover, in transformer-based symbolic regression, data memorization is desirable, because it directly improves recovery.

**Standardization is necessary.** In application areas of symbolic regression, for instance, physics, data are measured at vastly different scales and under different sampling conditions (Keren et al., 2023). Deep learning methods, however, are known to be highly susceptible to overfitting to the training domain and the sampling conditions. Therefore, standardization is necessary for transformer-based symbolic regression to avoid overfitting to the training domain, that is, to the domain from which the training data points are sampled.

**Scaling works.** While we were able to significantly boost recovery in transformer-based symbolic regression, we still do not achieve state-of-the-art performance. However, our experimental setup was limited to a single NVIDIA RTX A6000 GPU. Still, our experimental results provide evidence that the recovery performance of transformer-based symbolic regression scales with the size of the training dataset and with the number of model parameters. Since both the synthetic data generation method and the underlying model architecture are also scalable, we are confident that better results can be achieved with more resources.

## 4 Conclusion

The transformer-based approach to symbolic regression shifts computational effort from inference to training. Inference with transformer-based symbolic regressors is up to three orders of magnitude more efficient than competing search-based state-of-the-art approaches. Therefore, invested computational resources for training a transformer-based symbolic regressor more than amortize when they are reused for inference, for instance, by distributing the trained regressor to users or by providing inference as a service. However, hitherto the transformer-based approach was by far not able to compete with state-of-the-art regressors in terms of recovery, an important measure of interpretability. In this work, we have shown that recovery in transformer-based symbolic regression can be boosted significantly by using a carefully designed training dataset. In our experiments, using fairly limited computational resources, we have not yet reached the recovery performance of state-of-the-art regressors. We have shown, however, that recovery scales favorably with the available computational resources. Thus, it seems likely that state-of-the-art performance can be achieved with larger computational budgets.

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
