# Boosting Recovery in Transformer-Based Symbolic Regression (Supplementary Material)

submitted to ICLR 25

## Contents

# 1  Details on Training Data Generation

This section contains details of the training data generation procedure. The Section 1.1 presents the parameters used for sampling the training RMPs. Section 1.2 covers implementation details of the RMP selection algorithm. Section 1.3 discusses combinatorial properties of RMP enumeration and feasibility issues. Section 1.4 focuses on details of the extended RMP selection method using RMP recombination.

## 1.1  Parameters

All parameters used for the training data generation are listed in Table 1.

Table 1: Training data generation parameters.

| Parameter | Description | Value |
|---|---|---|
| $I$ | Instruction Set | `add, mult, inv, neg, sin, cos, log,`  `exp, sq, sqrt` |
| $I_1$ | Unary Instructions | `inv, neg, sin, cos, log, exp, sq,`  `sqrt` |
| $I_2$ | Binary Instructions | `add, mult` |
| $D$ | Number of Input Dimensions | $d \in \{1, \ldots, 10\}$ |
| $L$ | Program Lengths | $l \in \{1, \ldots, 12\}$ |
| $A$ | Sampling Range | $a \in [-10, 10]$ |
| $N$ | Number of Sample Points | $n \in \{192, 448, 960\}$ |
| $B$ | Constant Range | $b \in [-10, 10]$ |
| $D_{max}$ | Max Input Dimension | 10 |
| $L_{max}$ | Max RMP Length | 12 |

## 1.2  Implementation Details

The detailed RMP selection procedure for training data generation consists of the following steps:

1. **Enumeration**

   (a) **Hyperparameters.** Determine the input dimension $D \in \{1, \ldots, D_{max}\}$ and the maximum length $L \in \{1, \ldots, L_{max}\}$ of RMPs that are feasible to enumerate within the constraints of the available computational budget. In our case, these are programs with a maximum length of 5 lines, thus allowing a maximum of 6 input variables.

   (b) **Enumeration.** Generate all $|I|^L$ instruction sequences as skeletons for the RMPs. Turn each RMP skeleton into a list of valid RMPs by generating all valid combinations of registers for all instruction steps of the RMP. In the corresponding DAG, this is equivalent to all the possible wirings of the DAG with the given number of operators. The result is a list of all enumerable, valid RMPs for the given length.

(c) **Equivalence Check.** Generate a verification set of $N$ random input points $x \in \{-a, a\}^{D_{max}}$ and store it as the base for the following hash generation procedure. We set $N = 1000$ and $a = 10$ in our case. Infer all RMPs on this verification set. Check if at least 10% of the points are valid/non-NaN values. If not, the program is rejected. Standardize the resulting output array $\{y_i\}_{i=1}^{N}$ of each RMP by subtracting the mean and dividing by the standard deviation. Then hash all standardized output arrays and store the hashes in a hash table. In the event of a hash collision, keep only the RMP with the shorter length. In the case of equal length, add the RMP to the list of minimal alternatives. This effectively compresses the list of enumerated RMPs to the minimal RMPs, including alternatives.

2. **Recombination**

(a) **Hyperparameters.** Set an upper bound on the length $L$ of RMPs to be generated from combinations. In our case, we generate programs up to a length of $L = 12$.

(b) **Recombination.** Randomly select two RMPs from the enumerated, compressed set of minimum RMPs from step 1. Connect the output of the first RMP to a randomly selected input variable of the second RMP. Then randomly select variables from all the input variables $x_i$ of the first RMP, all the intermediate variables $a_i$ of the first RMP, and the remaining input variables $x_j$ of the second RMP and connect them to the inputs of the second RMP until all the inputs of the second RMP are connected. The result is a new, larger RMP that is considered a candidate for the dataset. To be included in the data set, the candidate undergoes a series of checks.

3. **Verification**

(a) **Common Subexpression Check.** Eliminate RMPs that are longer than the specified maximum length, $L$. Also, reject candidates that contain common subexpressions (CSEs). For these RMPs, we know that shorter representations must exist.

(b) **Gradient Check.** Infer all candidates on the verification set of $N$ random points generated in step 1. For each intermediate variable $a_i$ in the RMP, trace back the input variables that influence it. Then, for each intermediate variable, check that the gradients of all influencing input variables are non-zero. This ensures that the instruction at this step does not lead to the elimination of input variables, as in $x/x = 1$ or $\sin^2(x) + \cos^2(x) = 1$. Furthermore, we check if the output variable $a_i$ receives gradients from all input and intermediate variables. Otherwise, we know that an intermediate variable does not contribute to the program and is therefore redundant, and a shorter representation must exist.

(c) **Equivalence Check.** Standardize the output array $\{y_i\}_{i=1}^{N}$ by subtracting the mean and dividing by the standard deviation. Then hash it and store the hash in the hash table. In case of collisions, keep only the RMP with the shorter length. In the case of equal length, add the RMP to the list of minimal alternatives.

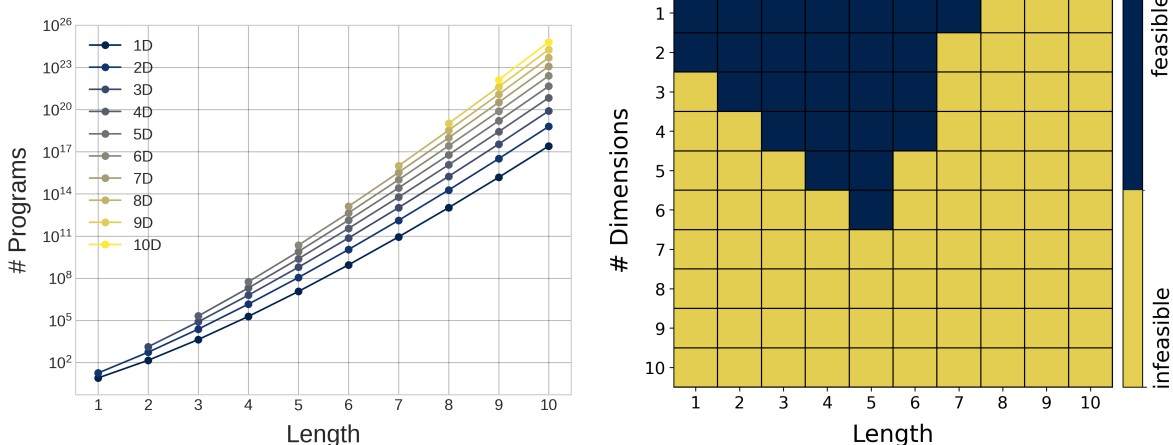

Figure 1: **Enumeration and verifiability of RMPs.** *Left:* Total number of programs per length and dimension (y-axis is log scale). *Right:* Feasible number of RMPs distributed across $(D, L)$ buckets that can be checked within a given computation budget of $10^9$ equivalence checks.

## 1.3 Register Machine Program Enumeration

**Feasibility issues.** Register machine program enumeration allows to find exhaustively minimal RMPs by enumerating them and then filtering them by efficient equivalence checking. However, due to the exponential growth of the space of possible RMPs over their length, the range that can be effectively enumerated in our computational budget is limited. The left part of Figure 1 illustrates the large increase in the number of programs as program length increases. The right part of Figure 1 illustrates the small number of programs that can be feasibly enumerated versus the large number of $(D, L)$ buckets that cannot be feasibly enumerated. A $(D, L)$ bucket is colored in dark blue if the number of programs is below the given computational budget of $10^9$ equivalence checks. In practice, we try to maximize the **total** number of buckets that can be enumerated within a computational budget of $10^9$ equivalence checks. This includes all RMPs with a maximum length of 5 lines and a maximum of 5 inputs and takes approximately 1 hour.

**Growth behavior of RMPs.** Within the feasibly enumerable range of programs ($D <= 5, L <= 5$), we compare the total number of enumerable programs with the number of equivalence classes reached by these programs, as well as the total number of minimal RMP alternatives. As equivalence classes, we count the number of different unique hashes of the outputs of all enumerated programs. As minimal RMPs, we count all RMPs of the same length that produce the same output, including all minimal alternatives. The left part of the Figure 2 shows the results of the comparison. It can be seen that the number of equivalence classes, while still growing exponentially (y-axis is log scale), grows much slower than the total number of programs. We can also see that the number of minimal RMPs including alternatives grows at about the same rate as the number of equivalence classes. This shows that for a single equivalence class there are only a small number of minimal programs that generate it.

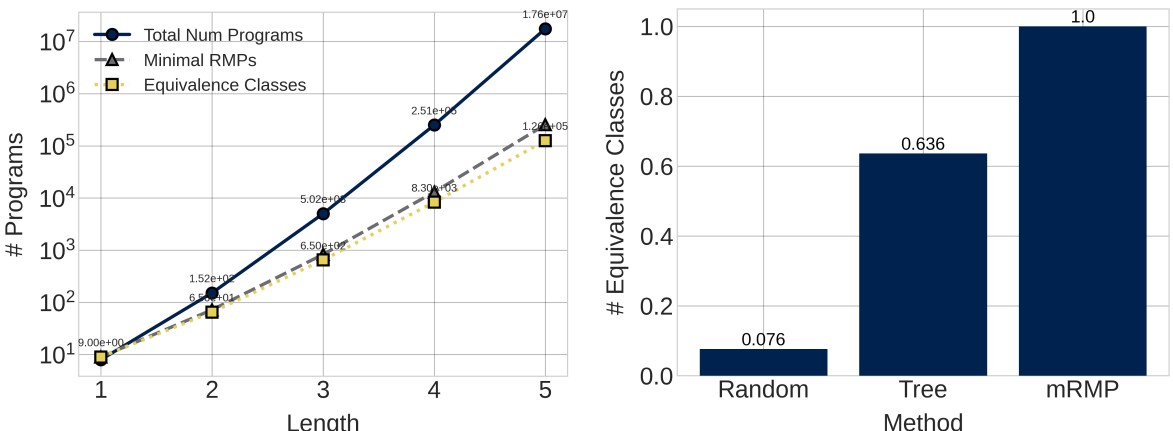

Figure 2: **Total number of programs vs. equivalence classes.** *Left:* Relation of total number of programs to minimal RMPs. The total number of programs grows several orders of magnitude faster than the number of equivalence classes (y-axis is in logarithmic scale). For a given equivalence class, there may be several minimal RMP alternatives. *Right:* Comparison of random RMP generation, expression tree generation after Kamienny *et al.* [2022], and minimal RMP generation in terms of the number of different equivalence classes reached by each training data generation method after taking 10k samples from each. The values presented have been normalized.

**Comparing training data generation methods.** To evaluate the quality of our training data generation method with other synthetic data generation methods, we place it in the context of pure random RMP instruction sequence generation and the expression tree generation method used by Kamienny *et al.* [2022]. For all three methods, we use the same instruction set $I$ as listed in Table 1. First, we compute a test set of $N = 1000$ random input points $x \in \mathbb{R}^D$ in the range $[-10, 10]^D$. Then we draw 10k samples from each training data generation method, resulting in 10k expressions for each method. We then execute the expressions as lambda functions on the test input points and hash the output arrays $\{y_i\}_{i=1}^N$. Then we determine the number of different equivalence classes reached by each method. This is done by counting the unique hashes reached by each approach. The right part of the Figure 2 shows the results of the comparison. Random RMP generation within 10k samples achieves only 760 different equivalence classes. This is due to the fact that without any restriction on the program structure, many generated programs lead to poorly connected instruction sequences, where only a subset of the input variables affect the output. The random expression tree generation method imposes more restrictions on the structure of the expression tree, leading to more diverse expressions even for a higher number of input variables and length. However, without explicitly optimizing the procedure to achieve a high diversity of equivalence classes, it only achieves about 6360 different equivalence classes on 10k samples. Another problem with both approaches is that, because there are no implemented checks on the structure of the generated expressions, we have no confidence that the generated programs, even if they reach different equivalence classes, are actually guaranteed to be compact representations. Our method is specifically optimized to sample only minimal RMPs from different equivalence classes. When

10k samples are taken, the generation algorithm is designed to return only RMPs that reach different equivalence classes.

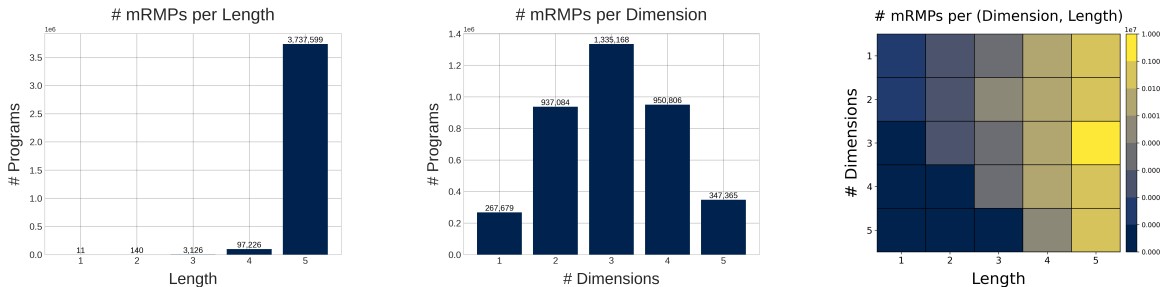

Figure 3: **Statistics of verified minimal RMPs.** *Left:* Number of minimal RMPs per length. *Center:* Number of minimal RMPs per input dimension. *Right:* Distribution of minimal RMPs over different input dimension-length buckets.

**Statistics of the enumerable RMPs.** Figure 3 shows the statistics of the minimal RMPs generated by filtering the enumerated set of all RMPs in $(D <= 5, L <= 5)$. We can see that, as expected, longer programs contain exponentially more minimal RMPs than shorter ones.

## 1.4   Extended Register Machine Program Selection

We extend our RMP selection method with a recombination step to generate RMPs with compact representations for buckets with $(D > 5, L > 5)$. RMP recombination allows to efficiently generate good candidates for compact programs from existing verified minimal RMPs.

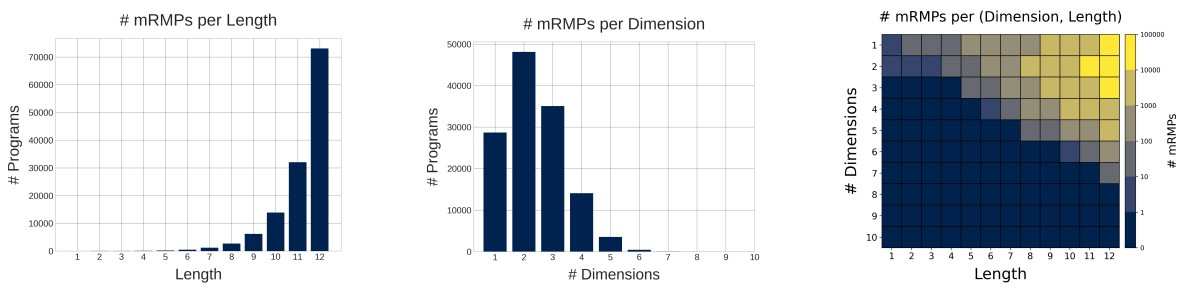

Figure 4: **Statistics of recombination from initial instruction set.** Statistics of the minimal RMPs generated by random recombination starting with the basic instruction set $I$ as initial set functions. Only the programs with the shortest length after $10^9$ equivalence checks are kept.

As an absolute base case, we illustrate how the combinatorial process evolves when only the basic instructions in $I$ are set as the initial set of programs for recombination. In practice, this means that we start with all unary instructions as RMPs of length one with input dimension one, and all binary instructions as RMPs of length one with input dimension two. Figure 4

shows the distribution of minimal RMPs over the length, dimension, and $(D, L)$ buckets. We can see that when programs are randomly recombined from the initial instruction set without restrictions, the programs rapidly evolve towards programs of larger length.

To mitigate this behavior and ensure a recombination process that generates programs distributed over all $(D, L)$ buckets equally, we make a few changes to the base case above. First, we set the initial set for recombination to the minimal RMPs enumerated and verified in step 1 of the training data generation procedure. Then, when recombining, we sample programs for recombination uniformly at random across all $(D, L)$ buckets. This results in the generation of novel, compact RMPs that are roughly evenly distributed across all $(D, L)$ buckets, forming a balanced training data set. Figure 5 shows the resulting statistics of the RMPs obtained from recombination step 2.

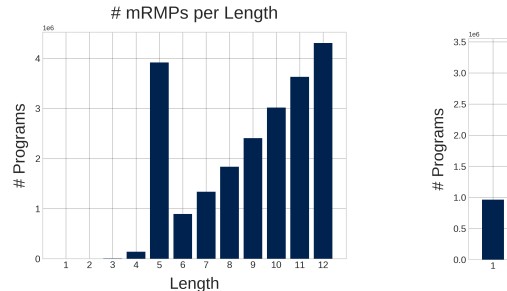 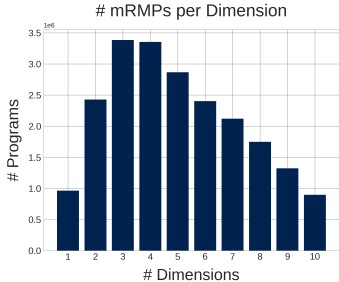 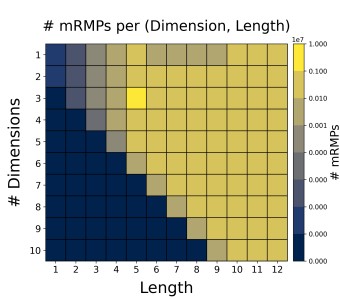

Figure 5: **Statistics of recombination from minimal RMPs as seed set.** Statistics of the set of RMPs generated by random recombination from an initial set of minimal RMPs obtained from step one of the data generation process. The algorithm is stopped after 16M equivalence classes are reached.

## 2 Details on Model Training

The Table 2 contains the hyperparameters used to train the transformer model. The model is trained by minimizing a cross-entropy loss using the Adam optimizer. The learning rate is initially set to $10^{-7}$ and then, over the first 2000 steps, warmed up to $10^{-4}$. Subsequently, a cosine learning rate decay to $10^{-5}$ is employed, as in Lewkowycz [2021].

## 3 Details on Experiments

This section presents additional details on the results of the experiments. Section 3.1 discusses additional results on metrics other than recovery on the SRBench benchmark. Section 3.2 presents details on the ablation studies, specifically looking at recovery performance over certain program lengths, dimensions, and (D,L) buckets, as well as providing additional metrics on the Feynman dataset.

Table 2: Model training hyperparameters.

| Parameter | Value |
|---|:---:|
| Embedding Dimension | 512 |
| Num Encoder Layers | 4 |
| Encoder Block Size | {192, 448, 960} |
| Num Decoder Layers | {1, 2, 4, 8, 16} |
| Decoder Block Size | 64 |
| Num Attention Heads | 16 |
| Batch Size | 32 |
| Learning Rate | $10^{-4}$ |
| Warmup Iterations | 2000 |
| Weight Decay | $10^{-1}$ |
| Adam Betas | $(0.9, 0.95)$ |
| Dropout | 0.0 |
| Grad Clip | 0.5 |

## 3.1 Additional SRBench Results

The SRBench test suite provides insight into other metrics besides recovery. Kahlmeyer *et al.* [2024] suggest using the Jaccard index metric as a non-binary measure of recovery to illustrate how many components of a formula were correctly recovered by a symbolic regressor. The left part of the Figure 6 shows the result of this evaluation. Our E2E-RMP approach ranks second in this evaluation, indicating a high recovery of correct components.

Looking at model fit metrics such as the $R^2$-score, we can see that our E2E-RMP approach ranks lower than the E2E approach. This can be explained by the fact that we restrict our RMPs to contain at most 1 constant in order to obtain very compact program representations. Other methods do not have this limitation and are therefore allowed to include an unlimited number of constants in the functions. This gives the generated candidates a much higher degree of freedom than the programs generated by our E2E-RMP regressor and explains their higher $R^2$-scores on the SRBench benchmark. Nevertheless, our method is deliberately designed with a recovery-first approach in mind, and therefore omits additional complexity in the form of constants wherever possible. The trade-off between the fit of the generated expression and its complexity is illustrated in Figure 7.

Table 3 compares the recovery rates and the Jaccard index as the main metrics of recovery, as well as the model fit metric $R^2$ and the expression complexity between transformer-based end-to-end symbolic regression approaches. The influence of the synthetic training data in the form of compact RMPs can be clearly seen in the recovery and Jaccard index metrics, representing a large improvement on these metrics.

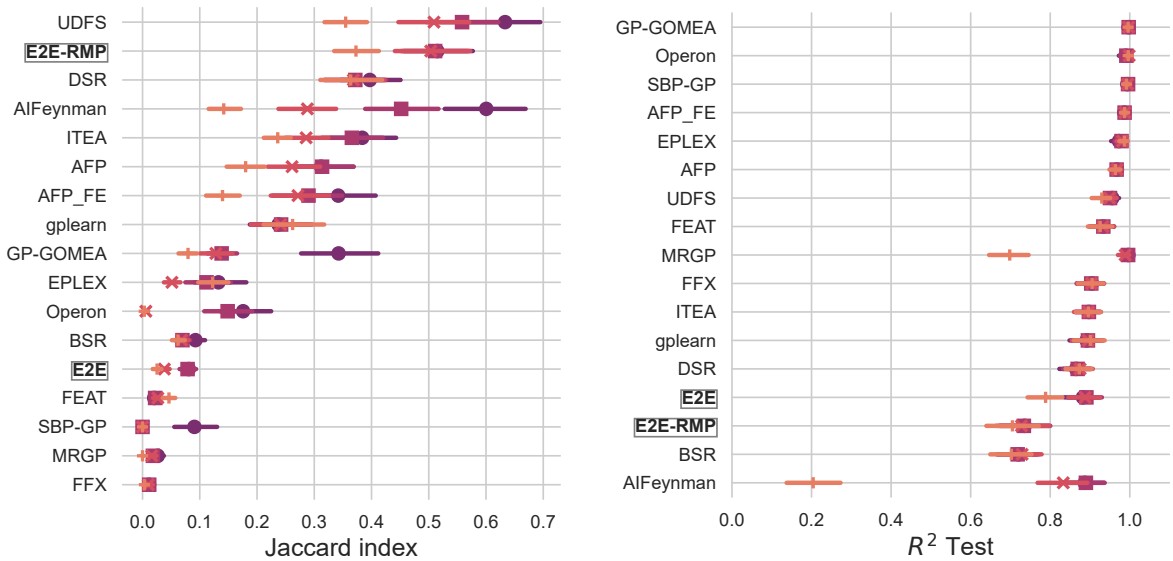

Figure 6: **SRBench model fit evaluation.** *Left:* Average Jaccard index, which measures the overlap of the predicted formulas with the ground truth formulas. *Right:* Average $R^2$ score on the SRBench benchmark. The reported values are averages over 10 runs.

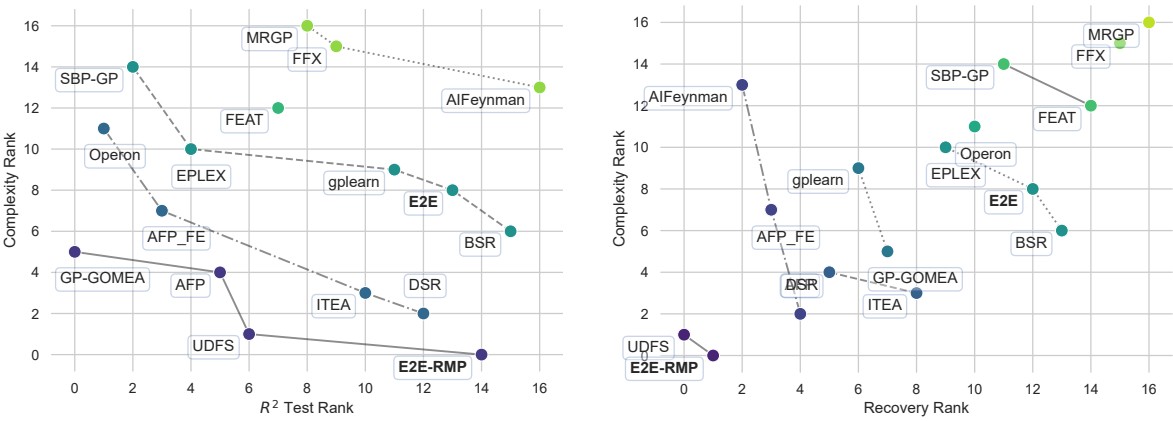

Figure 7: **SRBench pareto evaluation.** *Left:* Model fit to model complexity tradeoff for all compared methods. *Right:* Recovery to model complexity trade-off for all methods compared.

## 3.2 Details on Ablation Studies

In Table 4, we present additional results for the scaling experiments over the parameter size of the model. The parameter size is increased by scaling the number of decoder layers from one to 16. We report recovery, $R^2$-score, Root Mean Squared Error (RMSE), and complexity on the Feynman data set [Udrescu and Tegmark, 2020b]. In addition, we report recovery performance on the training and test sets. Larger parameter sizes consistently show higher recovery rates.

Table 3: **Comparison of transformer-based end-to-end approaches on SRBench and reduced Feynman dataset.** The upper part of the table shows the recovery results of transformer-based approaches on SRBench. The transformer approach trained on minimal RMPs shows higher overall recovery and higher overlap with ground truth formulas (Jaccard index), as well as low formula complexity. In the middle of the table, the performance of the transformer-based approaches are compared with state-of-the-art approaches that are not transformer-based. At the bottom of the table, we separately compare the E2E-RMP approach with the transformer of Biggio *et al.* [2021], because Biggio *et al.* [2021] only allows evaluation on formulas with an input dimension $< 4$. We therefore evaluated the approach on the 48 possible formulas out of the 119 formulas in the Feynman dataset and compared it with our approach on the same dataset. We compare our $86M$ variant with the $100M$ variant of Biggio *et al.* [2021]. First, we compare both approaches on points sampled in-domain in $[-10, 10]$, which we highlight with ID in the table. The E2E-RMP transformer has a seven per cent higher recovery rate on this split of the data. We explain this effect by the systematic generation of minimal RMPs for training. Then, we compared the out-of-domain performance by moving the domain progressively to the right, starting from ten up to 40. What we observe is a decrease in recovery performance for both methods, although E2E-RMP degrades much more gracefully, which we attribute to standardization. Out-of-domain performance results are marked OOD.

| Method | Recovery in % | Jaccard Index | R2 | Complexity |
|---|---|---|---|---|
| **TF-approaches on SRBench** | | | | |
| Kamienny *et al.* [2022] | 1.59 | 0.08 | 0.88 | 31.95 |
| Vastl *et al.* [2022] | 2.87 | 0.12 | 0.43 | 17.63 |
| Lalande *et al.* [2023] | 0.00 | 0.14 | 0.05 | 8.36 |
| Shojaee *et al.* [2023] | 33.89 | 0.29 | 0.99 | 35.74 |
| E2E-RMP | 34.02 | 0.51 | 0.72 | 11.22 |
| **Non TF-approaches on SRBench** | | | | |
| Cranmer [2023] | 44.73 | 0.58 | 0.97 | 13.15 |
| **TF-approaches on reduced Feynman dataset** | | | | |
| Biggio *et al.* [2021]-ID | 47.92 | 0.59 | 0.85 | 8.27 |
| E2E-RMP-ID | 54.67 | 0.55 | 0.87 | 9.89 |
| Biggio *et al.* [2021]-OOD | 4.16 | 0.19 | 0.29 | 13.60 |
| E2E-RMP-OOD | 14.58 | 0.26 | 0.09 | 8.21 |

For the largest model variant with 16 decoder layers, we perform ablations over dataset size, beam size, and input length. In the dataset size experiments, we scale the number of equivalence classes a model sees during training from one to 16. We train each model until the validation loss is saturated and then measure the recovery performance. Regardless of the number of model parameters, we observe a stable increase in recovery performance as more equivalence classes are seen during training. Given the synthetic nature of the data generation method, we

Table 4: **Detailed recovery results on the Feynman dataset.** Comparison of the recovery performance of the transformer model with different numbers of decoder layers on the well-known Feynman dataset, the training set, and the test set.

| Model | Recov Feynman | R2 | RMSE | Complexity | Recov Test | Recov Train |
|-------|---------------|------|------|------------|------------|-------------|
| TF-1  | 0.40          | 0.83 | 0.09 | 10.7       | 0.09       | 0.11        |
| TF-2  | 0.40          | 0.86 | 0.09 | 11.0       | 0.10       | 0.11        |
| TF-4  | 0.43          | 0.85 | 0.07 | 11.4       | 0.10       | 0.13        |
| TF-8  | 0.47          | 0.85 | 0.06 | 11.4       | 0.12       | 0.14        |
| TF-16 | 0.49          | 0.87 | 0.07 | 11.2       | 0.13       | 0.15        |

expect the recovery performance to increase further as more synthetic data is generated. In the beam size experiments, we infer the model with varying beam sizes from one to 16. The larger the beam size, the better the recovery performance. In the input length experiments, we fine-tune model variants for encoder input sizes of 192, 448, and 960 tokens. As the number of tokens in the input increases, the recovery performance decreases, indicating an increase in the difficulty for the model. The performance degradation seems to slow down as the number of parameters increases, comparing the 8-layer and 16-layer variants. One potential explanation for this phenomenon is the increasing computational complexity of the task for a model with the same parameter size when presented with a greater number of inputs.

**Recovery per length and dimension.** Figure 8 illustrates the recovery performance of the model on the Feynman dataset over different program lengths and dimensions. It can be observed that programs with shorter lengths and programs with fewer input dimensions are generally recovered better than longer programs or programs with a larger number of inputs. This is not a phenomenon specific to the E2E architecture. Furthermore, the steady decline in recovery performance can be observed for various state-of-the-art symbolic regressors such as UDFS Kahlmeyer *et al.* [2024], DSR Landajuela *et al.* [2021] or PySR Cranmer [2023], as shown in Figure 9. Note that complexity here is usually measured as the size of the minimal expression tree of the target expression to be recovered.

**Recovery over dimension-length buckets.** Figure 10 illustrates the recovery performance of model ablations with different parameter sizes over all $(D, L)$ buckets. Models with a larger number of parameters consistently achieve higher recovery across all buckets.

**Inference speed comparison.** To put the inference speeds of the various state-of-the-art symbolic regression methods in the context of the end-to-end approach, we measure the performance of the E2E-RMP transformer on our setup with a single NVIDIA RTX A6000 GPU. Table 6 shows the results of the comparison. Compared to other methods, speedups of 2.3 to 3 orders of magnitude (depending on the inference platform) are achieved.

Table 5: **Detailed ablation study results on the Feynman dataset.** Applying different ablations to the dataset size, beam size, and input length to the 16-decoder layer variant of the transformer model. The bottom shows an additional ablation. It compares an 8 decoder layer variant at different input lengths.

| Ablation | Recov Feynman | R2 | RMSE | Complexity |
|---|---|---|---|---|
| TF-16 Dataset Size 1M | 0.34 | 0.79 | 0.07 | 11.72 |
| TF-16 Dataset Size 2M | 0.36 | 0.76 | 0.07 | 12.86 |
| TF-16 Dataset Size 4M | 0.38 | 0.79 | 0.07 | 12.06 |
| TF-16 Dataset Size 8M | 0.45 | 0.79 | 0.06 | 11.51 |
| TF-16 Dataset Size 16M | 0.50 | 0.88 | 0.07 | 11.30 |
| TF-16 Beam Size 1 | 0.21 | 0.56 | 0.05 | 17.68 |
| TF-16 Beam Size 2 | 0.45 | 0.76 | 0.05 | 12.28 |
| TF-16 Beam Size 4 | 0.48 | 0.80 | 0.04 | 11.52 |
| TF-16 Beam Size 8 | 0.48 | 0.84 | 0.07 | 11.16 |
| TF-16 Beam Size 16 | 0.51 | 0.91 | 0.06 | 10.97 |
| TF-16 Input Length 192 | 0.50 | 0.88 | 0.07 | 11.30 |
| TF-16 Input Length 448 | 0.43 | 0.77 | 0.05 | 12.42 |
| TF-16 Input Length 960 | 0.39 | 0.81 | 0.07 | 12.35 |
| TF-8 Input Length 192 | 0.45 | 0.88 | 0.07 | 11.66 |
| TF-8 Input Length 448 | 0.39 | 0.80 | 0.06 | 12.81 |
| TF-8 Input Length 960 | 0.33 | 0.82 | 0.04 | 13.12 |

**Detailed analysis of the formulas recovered from the Feynman data set.** The Feynman problems comprise formulas from the renowned Feynman Lectures Feynman *et al.* [2011] and were initially introduced by Udrescu and Tegmark [2020a] to assess the efficacy of their symbolic regressor. Subsequently, it has been employed and validated by La Cava *et al.* [2021] in their comprehensive benchmark suite. The database can be accessed via the following link: here.

| Name | Expression | Name | Expression |
|---|---|---|---|
| I.10.7 | $\dfrac{x_0}{\sqrt{-\frac{x_1^2}{x_2^2}+1}}$ | I.11.19 | $x_0 x_3 + x_1 x_4 + x_2 x_5$ |
| **I.12.1** | $x_0 x_1$ | **I.12.11** | $x_0\left(x_1 + x_2 x_3 \sin\left(x_4\right)\right)$ |
| **I.12.2** | $\frac{x_0 x_1}{4\pi x_2 x_3^2}$ | I.12.4 | $\frac{x_0}{4\pi x_1 x_2^2}$ |
| **I.12.5** | $x_0 x_1$ | I.13.12 | $x_0 x_1 x_4 \cdot \left(\frac{1}{x_3} - \frac{1}{x_2}\right)$ |
| **I.13.4**[*] | $\frac{x_0\left(x_1^2 + x_2^2 + x_3^2\right)}{2}$ | **I.14.3** | $x_0 x_1 x_2$ |
| **I.14.4** | $\frac{x_0 x_1^2}{2}$ | I.15.3t | $\dfrac{-\frac{x_0 x_2}{x_1} + x_3}{\sqrt{1 - \frac{x_2^2}{x_1^2}}}$ |

Continues on next page

| Name | Expression | Name | Expression |
|------|-----------|------|-----------|
| I.15.3x | $\dfrac{x_0 - x_1 x_3}{\sqrt{-\frac{x_1^2}{x_2^2} + 1}}$ | I.16.6 | $\dfrac{x_1 + x_2}{1 + \frac{x_1 x_2}{x_0^2}}$ |
| I.18.12 | $x_0 x_1 \sin(x_2)$ | I.18.14 | $x_0 x_1 x_2 \sin(x_3)$ |
| **I.18.4** | $\dfrac{x_0 x_2 + x_1 x_3}{x_0 + x_1}$ | I.24.6 | $\dfrac{x_0 x_3^2 (x_1^2 + x_2^2)}{4}$ |
| **I.25.13** | $\dfrac{x_0}{x_1}$ | I.27.6 | $\dfrac{1}{\frac{x_2}{x_1} + \frac{1}{x_0}}$ |
| I.29.16 | $\sqrt{x_0^2 - 2 x_0 x_1 \cos(x_2 - x_3) + x_1^2}$ | **I.29.4** | $\dfrac{x_0}{x_1}$ |
| I.30.3 | $\dfrac{x_0 \sin^2\left(\frac{x_1 x_2}{2}\right)}{\sin^2\left(\frac{x_1}{2}\right)}$ | I.32.17 | $\dfrac{4\pi x_0 x_1 x_2^2 x_3^2 x_4^4}{3\left(x_4^2 - x_5^2\right)^2}$ |
| **I.32.5**[*] | $\dfrac{x_0^2 x_1^2}{6\pi x_2 x_3^3}$ | I.34.1 | $\dfrac{x_2}{1 - \frac{x_1}{x_0}}$ |
| I.34.14 | $\dfrac{x_2 \cdot \left(1 + \frac{x_1}{x_0}\right)}{\sqrt{1 - \frac{x_1^2}{x_0^2}}}$ | **I.34.27** | $\dfrac{x_0 x_1}{2\pi}$ |
| **I.34.8** | $\dfrac{x_0 x_1 x_2}{x_3}$ | I.37.4 | $x_0 + x_1 + 2\sqrt{x_0 x_1}\cos(x_2)$ |
| **I.38.12**[*] | $\dfrac{x_2^2 x_3}{\pi x_0 x_1^2}$ | **I.39.1** | $\dfrac{3 x_0 x_1}{2}$ |
| I.39.11 | $\dfrac{x_1 x_2}{x_0 - 1}$ | **I.39.22** | $\dfrac{x_0 x_1 x_3}{x_2}$ |
| I.40.1 | $x_0 e^{-\frac{x_1 x_2 x_4}{x_3 x_5}}$ | I.41.16 | $\dfrac{x_0^3 x_2}{2\pi^3 x_4^2 \left(e^{\frac{x_0 x_2}{2\pi x_1 x_3}} - 1\right)}$ |
| **I.43.16** | $\dfrac{x_0 x_1 x_2}{x_3}$ | **I.43.31** | $x_0 x_1 x_2$ |
| I.43.43 | $\dfrac{x_1 x_3}{x_2 (x_0 - 1)}$ | **I.44.4**[*] | $x_0 x_1 x_2 \log\left(\frac{x_4}{x_3}\right)$ |
| **I.47.23** | $\sqrt{\dfrac{x_0 x_1}{x_2}}$ | I.50.26 | $x_0 \left(x_3 \cos^2(x_1 x_2) + \cos(x_1 x_2)\right)$ |
| I.6.2 | $\dfrac{\sqrt{2} e^{-\frac{x_1^2}{2 x_0^2}}}{2\sqrt{\pi} x_0}$ | I.6.2a | $\dfrac{\sqrt{2} e^{-\frac{x_0^2}{2}}}{2\sqrt{\pi}}$ |
| I.6.2b | $\dfrac{\sqrt{2} e^{-\frac{(x_1 - x_2)^2}{2 x_0^2}}}{2\sqrt{\pi} x_0}$ | **I.8.14**[*] | $\sqrt{(-x_0 + x_1)^2 + (-x_2 + x_3)^2}$ |
| I.9.18 | $\dfrac{x_0 x_1 x_2}{(-x_3 + x_4)^2 + (-x_5 + x_6)^2 + (-x_7 + x_8)^2}$ | **II.10.9** | $\dfrac{x_0}{x_1 (x_2 + 1)}$ |
| **II.11.20**[*] | $\dfrac{x_0 x_1^2 x_2}{3 x_3 x_4}$ | II.11.27 | $\dfrac{x_0 x_1 x_2 x_3}{-\frac{x_0 x_1}{3} + 1}$ |
| II.11.28 | $\dfrac{x_0 x_1}{-\frac{x_0 x_1}{3} + 1} + 1$ | II.11.3 | $\dfrac{x_0 x_1}{x_2 (x_3^2 - x_4^2)}$ |
| **II.13.17** | $\dfrac{x_2}{2\pi x_0 x_1^2 x_3}$ | II.13.23 | $\dfrac{x_0}{\sqrt{-\frac{x_1^2}{x_2^2} + 1}}$ |
| II.13.34 | $\dfrac{x_0 x_1}{\sqrt{-\frac{x_1^2}{x_2^2} + 1}}$ | **II.15.4** | $-x_0 x_1 \cos(x_2)$ |
| **II.15.5** | $-x_0 x_1 \cos(x_2)$ | **II.2.42**[*] | $\dfrac{x_0 x_3 (-x_1 + x_2)}{x_4}$ |
| II.21.32 | $\dfrac{x_0}{4\pi x_1 x_2 \left(-\frac{x_3}{x_4} + 1\right)}$ | II.24.17 | $\sqrt{\dfrac{x_0^2}{x_1^2} - \dfrac{\pi^2}{x_2^2}}$ |
| **II.27.16** | $x_0 x_1 x_2^2$ | **II.27.18** | $x_0 x_1^2$ |
| **II.3.24** | $\dfrac{x_0}{4\pi x_1^2}$ | **II.34.11** | $\dfrac{x_0 x_1 x_2}{2 x_3}$ |
| **II.34.2** | $\dfrac{x_0 x_1 x_2}{2}$ | **II.34.29a** | $\dfrac{x_0 x_1}{4\pi x_2}$ |

| Name | Expression | Name | Expression |
|------|-----------|------|-----------|
| **II.34.29b**[*] | $\frac{2\pi x_0 x_2 x_3 x_4}{x_1}$ | **II.34.2a** | $\frac{x_0 x_1}{2\pi x_2}$ |
| II.35.18 | $\frac{x_0}{e^{\frac{x_3 x_4}{x_1 x_2}}+e^{-\frac{x_3 x_4}{x_1 x_2}}}$ | II.35.21 | $x_0 x_1 \tanh\left(\frac{x_1 x_2}{x_3 x_4}\right)$ |
| II.36.38 | $\frac{x_0 x_1}{x_2 x_3}+\frac{x_0 x_4 x_7}{x_2 x_3 x_5 x_6^2}$ | **II.37.1** | $x_0 x_1 (x_2+1)$ |
| II.38.14 | $\frac{x_0}{2x_1+2}$ | **II.38.3** | $\frac{x_0 x_1 x_3}{x_2}$ |
| **II.4.23** | $\frac{x_0}{4\pi x_1 x_2}$ | II.6.11 | $\frac{x_1 \cos(x_2)}{4\pi x_0 x_3^2}$ |
| II.6.15a | $\frac{3x_1 x_5 \sqrt{x_3^2+x_4^2}}{4\pi x_0 x_2^5}$ | II.6.15b | $\frac{3x_1 \sin(x_2)\cos(x_2)}{4\pi x_0 x_3^3}$ |
| **II.8.31** | $\frac{x_0 x_1^2}{2}$ | **II.8.7** | $\frac{3x_0^2}{20\pi x_1 x_2}$ |
| III.10.19 | $x_0 \sqrt{x_1^2+x_2^2+x_3^2}$ | **III.12.43** | $\frac{x_0 x_1}{2\pi}$ |
| **III.13.18** | $\frac{4\pi x_0 x_1^2 x_2}{x_3}$ | III.14.14 | $x_0 \left(e^{\frac{x_1 x_2}{x_3 x_4}}-1\right)$ |
| III.15.12 | $2x_0 \cdot (1-\cos(x_1 x_2))$ | **III.15.14** | $\frac{x_0^2}{8\pi^2 x_1 x_2^2}$ |
| **III.15.27** | $\frac{2\pi x_0}{x_1 x_2}$ | III.17.37 | $x_0 (x_1 \cos(x_2)+1)$ |
| III.19.51 | $-\frac{x_0 x_1^4}{8x_2^2 x_3^2 x_4^2}$ | **III.21.20** | $-\frac{x_0 x_1 x_2}{x_3}$ |
| III.4.32 | $\frac{1}{e^{\frac{x_0 x_1}{2\pi x_2 x_3}}-1}$ | III.4.33 | $\frac{x_0 x_1}{2\pi\left(e^{\frac{x_0 x_1}{2\pi x_2 x_3}}-1\right)}$ |
| **III.7.38** | $\frac{4\pi x_0 x_1}{x_2}$ | III.8.54 | $\sin^2\left(\frac{2\pi x_0 x_1}{x_2}\right)$ |
| test 1 | $\frac{x_0^2 x_1^2 x_2^2 x_3^2 x_4^2}{16x_5^2 \sin^4\left(\frac{x_6}{2}\right)}$ | test 2 | $\frac{x_0 x_1\left(\sqrt{1+\frac{2x_2^2 x_3}{x_0 x_1^2}}\cos(x_4-x_5)+1\right)}{x_2^2}$ |
| test 3 | $\frac{x_0 \cdot (1-x_1^2)}{x_1 \cos(x_2-x_3)+1}$ | test 4 | $\sqrt{2}\sqrt{\frac{x_1-x_2-\frac{x_3^2}{2x_0 x_4^2}}{x_0}}$ |
| test 5 | $\frac{2\pi x_0^{\frac{3}{2}}}{\sqrt{x_1(x_2+x_3)}}$ | test 6 | $\sqrt{\frac{2x_0^2 x_1^2 x_6}{x_2 x_3^2 x_4^2 x_5^4}+1}$ |
| test 7 | $\sqrt{\frac{8\pi x_0 x_1}{3}-\frac{x_2 x_3^2}{x_4^2}}$ | test 8 | $\frac{x_0}{\frac{x_0 \cdot (1-\cos(x_3))}{x_1 x_2^2}+1}$ |
| test 9 | $-\frac{32x_0^4 x_2^2 x_3^2 (x_2+x_3)}{5x_1^5 x_4^5}$ | test 11 | $\frac{4x_0 \sin^2\left(\frac{x_1}{2}\right)\sin^2\left(\frac{x_2 x_3}{2}\right)}{x_1^2 \sin^2\left(\frac{x_2}{2}\right)}$ |
| test 12 | $\frac{x_0\left(-\frac{x_0 x_1^3 x_3}{\left(x_1^2-x_3^2\right)^2}+4\pi x_2 x_3 x_4\right)}{4\pi x_1^2 x_4}$ | test 13 | $\frac{x_0}{4\pi x_4 \sqrt{x_1^2-2x_1 x_2 \cos(x_3)+x_2^2}}$ |
| test 14 | $x_0 \left(-x_2+\frac{x_3^3 (x_4-1)}{x_2^2 (x_4+2)}\right)\cos(x_1)$ | test 15 | $\frac{x_2 \sqrt{1-\frac{x_1^2}{x_0^2}}}{1+\frac{x_1 \cos(x_3)}{x_0}}$ |
| test 16 | $x_3 x_5 + \sqrt{x_0^2 x_1^4+x_1^2 (x_2-x_3 x_4)^2}$ | test 17 | $\frac{x_0^2 x_1^2 x_4^2 \cdot\left(1+\frac{x_4 x_5}{x_3}\right)+x_2^2}{2x_0}$ |
| test 18 | $\frac{3\left(\frac{x_1 x_4^2}{x_2^2}+x_3^2\right)}{8\pi x_0}$ | test 19 | $-\frac{\frac{x_1 x_4^4}{x_2^2}+x_3^2 x_5^2 \cdot (1-2x_4)}{8\pi x_0}$ |

Table 7: **Feynman dataset.** The Feynman dataset contains 119 formulas from Feynman's lectures. Formulas successfully recovered by the model are shown in bold. In addition, successfully recovered formulas resulting from generalization are indicated by an asterisk.

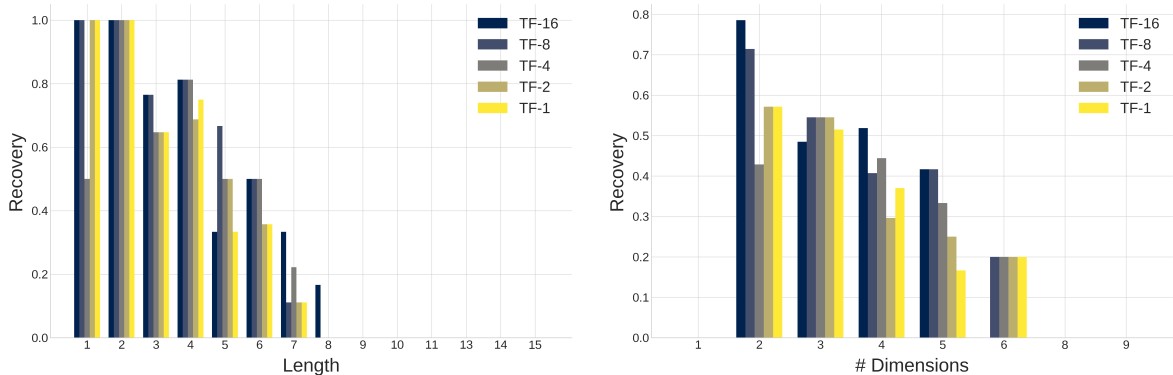

Figure 8: **Detailed recovery analysis on the Feynman dataset.** *Left:* Recovery results per program length on Feynman dataset. *Right:* Recovery results per input dimension on Feynman dataset.

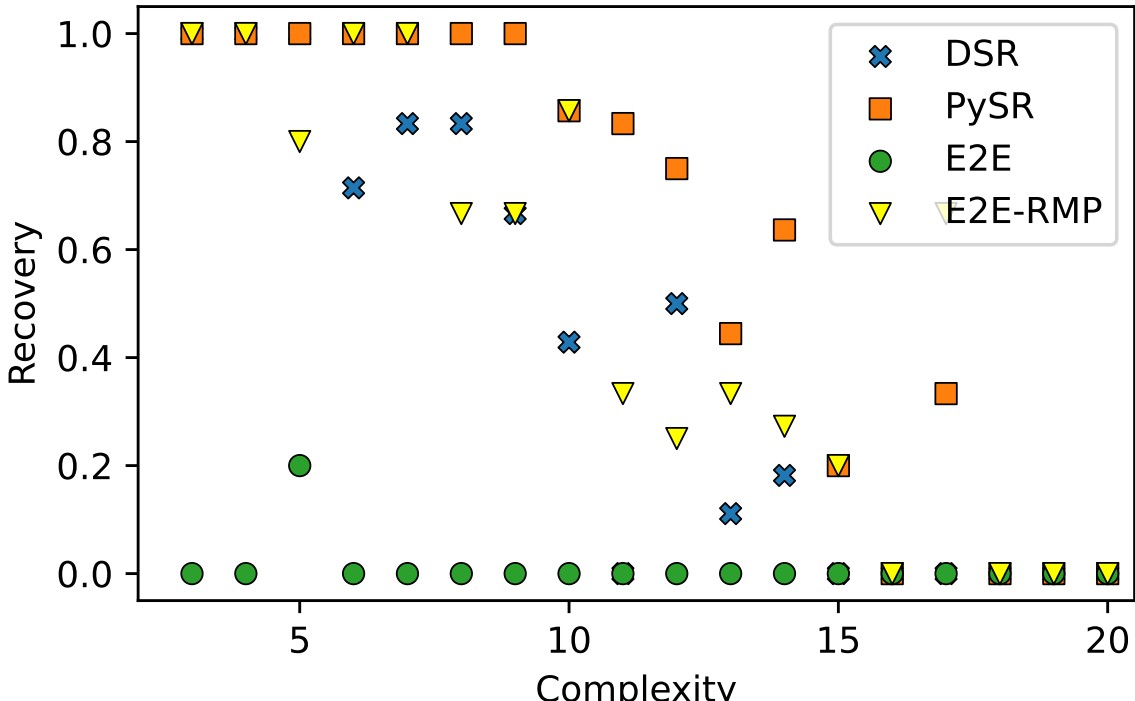

Figure 9: **Comparison of recovery over complexity for state-of-the-art symbolic regressors.** The recovery performance steadily decreases as the formulas to be recovered become more complex. This is a phenomenon observed for several state-of-the-art symbolic regressors.

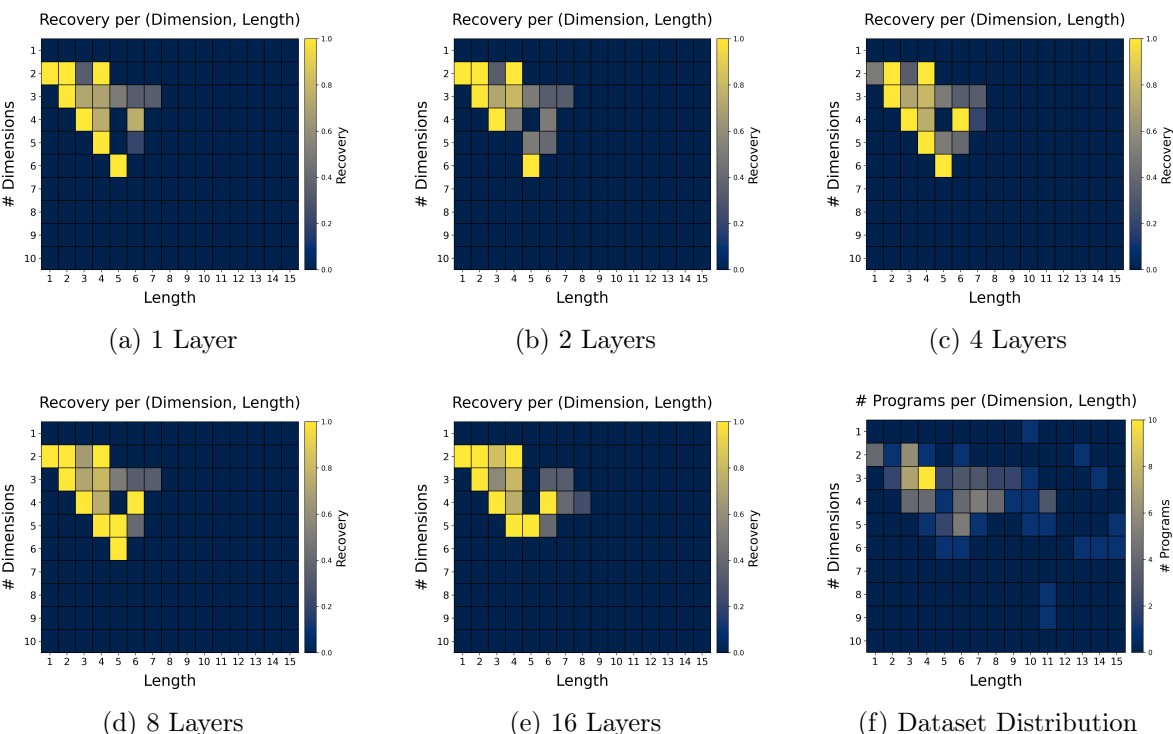

Figure 10: **Distribution of recovery results over** $(D, L)$ **buckets.** Distribution of recovery results for different model sizes over the different combinations of length and input dimension of the Feynman data set. The plot in the lower right-hand corner shows the overall distribution of the test programs in the data set.

Table 6: **Inference speed results.** Comparison of the inference speed of the transformer-based symbolic regression approach with state of the art search-based methods such as UDFS [Kahlmeyer *et al.*, 2024]. The average inference time for a single program is 2.3 orders of magnitude faster when inferred on an Intel(R) Xeon(R) Gold 6226R CPU @ 2.90GHz and 3 orders of magnitude faster when inferred on a single NVIDIA RTX A6000 GPU. The average times are calculated for a beam size of $k = 10$ over all programs of the Feynman dataset.

| Method | Avg. Time per Program (s) |
|---|---|
| DSR | 20.05 |
| gplearn | 9.10 |
| Operon | 7.28 |
| PySR | 3.82 |
| UDFS | 61.10 |
| TF-1 CPU | 0.32 |
| TF-2 CPU | 0.42 |
| TF-4 CPU | 0.59 |
| TF-8 CPU | 0.91 |
| TF-16 CPU | 1.58 |
| TF-1 GPU | 0.06 |
| TF-2 GPU | 0.08 |
| TF-4 GPU | 0.11 |
| TF-8 GPU | 0.17 |
| TF-16 GPU | 0.32 |