# OpenReview forum: "Boosting Recovery in Transformer-Based Symbolic Regression"
_ICLR.cc/2025/Conference — ICLR 2025 Conference Withdrawn Submission_

### Official Review · Reviewer_XjUE · 2024-11-02

**Soundness:** 2
**Presentation:** 1
**Contribution:** 2
**Rating:** 3
**Confidence:** 5

**Summary:**

The paper proposes a new generation method for end to end symbolic regression with transformers, replacing the tree representation commonly used to sample functions by Register Machine Programs (RMP), a representation equivalent to the directed acyclic graphs that appear when one removes common subexpressions in trees.

A training set is generated by enumerating all RMP for a given input dimension, up to a small number of instructions (ie operations). Then, the model proposed in Kamienny et al. (2022) is trained on this dataset, and tested on two small external test sets: Feynmann and Strogatz, for a total of 130 examples.

On this test set, the authors observe that the recovery rate of their new models is better than previous end to end transformer-based approaches.

**Strengths:**

The idea of adapting the training set of symbolic regression models, by selecting "more desirable", simpler, functions, is promising.

**Weaknesses:**

Novelty is extremely limited. The paper uses the architectures and generation techniques from D'Ascoli 2022 and Kamienny 2022, and the constant estimation methods from Biggio 2021. The main new element is the introduction of RMP, but its advantage over the expression trees used in previous works is not clear.

In previous works, expression trees are randomly generated, and a random tree usually has no common subexpressions (i.e. the corresponding DAG is the tree). As a result, the RMP introduces in this paper are most of the time equivalent to the trees they are supposed to replace. Besides, the RMP seems to result in longer sequences than enumerated trees. For instance, the expression $-sin(x_0)+0.3sin(x_0-x_1)$, given as an example in section 3.1, can be represented as a tree with no common subexpressions, and tokenized as 10 tokens. The corresponding RMP uses 31 tokens. What is the benefit?

The paper is difficult to read, it contains a number of incorrect and controversial statements (see below and questions), and does not clearly describe the generation techniques used, and the architecture ablation performed.  For example, figure 4 suggests data generation includes noise, this is not mentioned in sections 2.2 and 2.4. Section 3.1 describes the architecture used as an encoder-decoder model (as Kamienny), but line 332 states "Therefore, a training example with 192, 448, and 960 data points and 64 RMP tokens results in total context sizes of 256, 512, and 1024, respectively." which evokes a decoder-only (GPT-like) architecture, where input and output are concatenated. This notion of "context length" is repeated in the ablations.

The evaluation is very limited. The model is evaluated on 130 examples only. Figure 5 suggests that model performance (recovery) on its train and test set is around 15%, but it is 40% on the Feynman dataset. This may be due to the fact the most functions in the Feynman set are extremely simple. Besides the authors acknowledge that more than half of the Feynman test functions are already in the train set. This data contamination weakens the claims made in the paper.

**Questions:**

* l.123 The random sampling method for expression trees was introduced in Lample & Charton 2020 (Deep learning for symbolic mathematics)
* l.147 Kamienny 2022 clearly shows the benefit of estimating constants at inference, and fine-tuning them using BFGS, over predicting a "skeleton" with a special token replacing the constant. You seem to be using the latter, why?
* l. 191 Charton 2021 is not about symbolic regression, maybe cite D'Ascoli 2023 instead, who observe the overfitting
* l. 310 the three token representation for floats was introduced in Charton 2021
* l. 324: "As usual, this number is rounded to the next power of two, which here is 64." why round the output sequence length to a power of two in an encoder-decoder architecture?
* l. 330  "The RMP tokens are embedded into demb-dimensional space using a standard embedding layer. The
embeddings are then fed into a standard transformer decoder stack." Are you feeding the desired output as output of the decoder? This is not clear.
* l. 333 "total context sizes of 256, 512, and 1024, respectively": context size makes no sense in a decoder-only architecture
* l. 339  "Models are trained until the loss on the validation set is saturated." Can you explain what you mean by "saturated loss"?
* l. 347 the R2 score certainly predates La Cava, it is usually attributed to Pearson.

---

> ### Author Response · Authors · 2024-11-14
> **Rebuttal**
>
> Thank you for your review. We welcome the opportunity to clarify the novelty of our approach before addressing your specific questions and comments.
>
> **Novelty.**  In contrast to search-based approaches to symbolic regression, the transformer-approach needs training data. That is, the performance of the latter approach depends on the architecture as well as on the training data. The focus in previous work was mostly on the architecture. You are completely right, we keep the architecture of Kamienny 2022. However, we propose a novel scheme for generating the training data. Previous sampling approaches, for instance D’Ascoli 2022, have two major shortcomings: 1. They are not able to systematically cover expressions with succinct representations and by that miss many important candidate functions that therefore can mostly not be recovered by the symbolic regressor, and 2. they are more dependent on the sampling domain, that is, they do not generalize beyond that domain. This becomes apparent in a poor recovery rate. We address the first shortcoming by generating *minimal* register machine programs for our training expressions, and the second shortcoming by standardizing the input as well as the output of the functions used in training.
>
>
>
> ### **Questions and answers**
>
>
> **l.123 The random sampling method for expression trees was introduced in Lample & Charton 2020 (Deep learning for symbolic mathematics)**
>
> Thank you, we have corrected this.
>
>  **l.147 Kamienny 2022 clearly shows the benefit of estimating constants at inference, and fine-tuning them using BFGS, over predicting a "skeleton" with a special token replacing the constant. You seem to be using the latter, why?**
>
> Kamienny 2022 have to fit many constants, because they do not use standardization on inputs *and* outputs, and thus have to rescale the outputs for prediction, which includes fitting many constants. Since we standardize the inputs and outputs, we avoid the rescaling and have expressions with fewer constants. Using skeletons in this scenario has the advantage that a beam search can cover a more diverse set of candidate functions.
>
> **l. 191 Charton 2021 is not about symbolic regression, maybe cite D'Ascoli 2023 instead, who observe the overfitting**
>
> Thank you, we have changed this.
>
> **l. 310 the three token representation for floats was introduced in Charton 2021**
>
> Thank you, we have changed this.
>
>
> **l. 324: "As usual, this number is rounded to the next power of two, which here is 64." why round the output sequence length to a power of two in an encoder-decoder architecture?**
>
> Here, we were referring to the decoder block size during batch training. Rounding the output sequence length to a power of two in an encoder-decoder architecture *during training* is computationally advantageous in terms of memory access, leading to faster attention computation. To avoid confusion, we have changed this in the paper.
>
> **l. 330 "The RMP tokens are embedded into d_emb-dimensional space using a standard embedding layer. The embeddings are then fed into a standard transformer decoder stack." Are you feeding the desired output as output of the decoder? This is not clear.**
>
> Since we employ the same architecture as Kamienny 2022, the RMP tokens are first embedded and then fed into the decoder *during training*. At inference, the decoder is, of course, inferred in a step-by-step next token prediction manner.
>
>
> **l. 333 "total context sizes of 256, 512, and 1024, respectively": context size makes no sense in a decoder-only architecture.**
>
> For the training, in the ablations, we were referring to the sum of the encoder block size (192, 448, or 960 tokens) and the decoder block size (64 tokens) as context length. In the paper, we have corrected this confusing notation, and now refer to the encoder block size as input length.
>
>
> **l. 339 "Models are trained until the loss on the validation set is saturated." Can you explain what you mean by "saturated loss"?**
>
> We optimize the cross-entropy loss over the training set. Every $k$ steps, we compute the loss on a validation set. If the validation loss decreases, then we store the decreased validation loss. If the validation loss does not decrease for $k\cdot n$ steps, then we consider it to be saturated and stop the training.
>
>
> **l. 347 the R2 score certainly predates La Cava, it is usually attributed to Pearson.**
>
> We did not mean to attribute the R2 score to La Cava, but only to mention that it is used in their benchmark. We have clarified this.

---

> > ### Comment · Reviewer_XjUE · 2024-11-25
> >
> > Thanks you very much for your replies, which clarify a number of points. However, I still struggle with the novelty, the benefits of introducing RMP (which in most cases are strictly equivalent to the trees they replace, because common subexpressions are rare in random trees), and the very limited experimental setting (SRBench is good as a baseline, but it is way too small to serve as a validation/test set for new ideas). I am therefore keeping my rating.

---

> > > ### Author Response · Authors · 2024-11-25
> > >
> > > Thank you for your answer.
> > >
> > > **Common subexpressions are rare in random trees.**
> > >
> > > As we have pointed out, randomly sampling representations is not a good idea, because one misses too many equivalence classes of functions with small representations. Therefore, we propose to systematically enumerate all small representations by enumerating minimal RMPs. When you just enumerate small expression tress, that is, trees up to a certain size, then you will have many trees with common subexpressions. That is, enumerating minimal expression trees is inefficient.
> > >
> > > **SRBench is good as a baseline, but it is way too small to serve as a validation/test set for new ideas.**
> > >
> > > Why?

---

> ### Author Response · Authors · 2024-11-21
> **Feedback on Rebuttal?**
>
> Thank you again for reviewing our paper. As you have expressed a strong opinion, we would appreciate your feedback on our rebuttal of your arguments.

---

### Official Review · Reviewer_W4ae · 2024-11-02

**Soundness:** 3
**Presentation:** 3
**Contribution:** 3
**Rating:** 5
**Confidence:** 4

**Summary:**

This paper addresses the recovery performance limitation in transformer symbolic regression models by introducing a novel data generation approach using register machine programs (RMPs). After pre-training previous transformer SR models with the same architecture on this new data, the authors demonstrate significant improvement in recovery rates while maintaining fast inference times.

**Strengths:**

* Addresses a valid and current challenge in transformer-based symbolic regression methods
* Takes a novel approach by focusing on data generation improvements
* Shows great results in the improvement of recovery rates from the new data generation setting

**Weaknesses:**

**Major Concerns:**

* Overlook of literature: For example, [1] is the pioneering work in transformer SR which also similarly follows skeleton-based training (having placeholder parameter for constants in symbolic expressions).

* While the paper focuses on recovery rate as the interpretability metric, evaluation on semantic symbolic correctness metrics such as out-of-domain generalization and extrapolation would also be helpful as they are more flexible than recovery rate and can consider other possibilities such as mathematical approximations or equivalence.

* The paper's approach of normalizing outputs in addition to inputs may fundamentally alter the underlying function behavior and the mapping to the corresponding symbolic function. For example, If we normalize the output, the functions with same skeleton but different constant values may collapse which might lead to negligible impact of some symbolic terms. I do think that normalization might help transformer to have better memorization mostly for simple problems like Feynman. How do authors make sure that the correspondence between symbolic expressions and data observations are following original data behavior after normalization, particularly for more complex expressions?

* The main novelty is in data generation for transformer SR model training, specifically the representation of expressions as register machine programs (RMPs). Not enough evidence is provided justifying RMP over expressions as sequence (prefix notation). Additional experiments are needed:
  1. Ablation for performance with different data generation components
  2. Comparison with [1] which also generates expression skeletons with placeholder parameters. [1] has shown a better recovery rate than (kamienny et al., 2022) due to its simpler data generation setting and focus on lower-dimensional problems. I would be interested to see the comparison of your method with [1] on Feynman problems with d_max = 3.


* Concern on the reported results in Figure 4:
  1. I don't understand why robustness to noise improves this much compared to other baselines. There's no detail from authors about adding noise to the new training data. Why this happen?

  2. Limited comparison with recent SR models like uDSR [2], TPSR [3] and PySR [4]

*  It's not clear what are the main features in new data generation setting that lead to this recovery boost? Ablation analysis is needed on the data generation components. For example, RPM vs expression prefix notation, target scaling, RMP verification steps, etc.


**Minor Comments:**
* Introduction writing cold be improved. There should be more focus on the contributions of the work than motivation or examples for symbolic regression. Figures 1-2 could move to appendix.

* What beam size / inference sampling size were used for E2E and E2E-RMP results?

---

[1] Biggio et al., Neural Symbolic Regression that Scales, 2021

[2] Landajuela et al., A Unified Framework for Deep Symbolic Regression, 2022

[3] Shojaee et al., Transformer-based Planning for Symbolic Regression, 2023

[4] Cranmer et al., Interpretable Machine Learning for Science with PySR and SymbolicRegression.jl, 2023

**Questions:**

provided above

---

> ### Author Response · Authors · 2024-11-15
> **Rebuttal**
>
> Thank you for your review. In the following, we address your major concerns.
>
> **Overlook of literature: [1] is the pioneering work in transformer SR which also follows skeleton-based training.**
>
> Thank you. This is indeed an unfortunate omission that we have corrected. We now give proper credit to this work.
>
> **While the paper focuses on recovery rate as the interpretability metric, evaluation on semantic symbolic correctness metrics such as out-of-domain generalization and extrapolation would also be helpful as they are more flexible than recovery rate and can consider other possibilities such as mathematical approximations or equivalence.**
>
> We agree, out-of-domain generalization and extrapolation are important objectives in regression. These objectives, however, are often well met by standard, non-symbolic regression methods. As we argue in the introduction, symbolic regression adds another objective, namely, interpretability, which can be assessed by the recovery performance of symbolic regressors. In the noise-free setting, recovery implies perfect out-of-domain generalization and extrapolation. In general, one should explore the accuracy-complexity Pareto front to trade off the two objectives, generalizability and interpretability.
>
> **How do authors make sure that the correspondence between symbolic expressions and data observations are following original data behavior after normalization?**
>
> The standardization practically affects only the shift and the scale of the output. In practice, those effects can be effectively dealt with by optionally fitting a constant to the generated expression.
>
> **Not enough evidence is provided justifying RMP over expressions as sequence (prefix notation). Additional experiments are needed:**
> - **Ablation for performance with different data generation components**
>
> For training the transformer, it is not important if the expressions are represented by RMPs or by expression trees in prefix notation, as long as they are *minimal*. We use RMPs because we can efficiently enumerate minimal RMPs up to a certain length. Another, though minor, advantage of  RMPs is that they can be easily sequentialized and therefore are well suited for the transformer architecture.
>
> - **I would be interested to see the comparison of your method with [1] on Feynman problems with d_max = 3.**
>
> We now have compared the method in [1] to our approach on Feynman problems with d < 4. The results are provided in the supplement. The results show that (1) in-domain E2E-RMP has a recovery rate that is ~7% better than [1], which we explain by the systematic enumeration of equivalence classes of functions with succinct representatives (minimal RMPS), and (2) out-of-domain E2E-RMP degrades significantly more gracefully than [1], which we explain by the in- and output standardization. Note also that, in contrast to us, [1] does contain all Feyman problems in their training set.
>
> **Concern on the reported results in Figure 4:**
> - **I don't understand why robustness to noise improves this much compared to other baselines. Why this happen?**
>
> Remember that we keep the architecture from Kamienny et al. (2022). To make their models more robust against noise, Kamienny et al. add different levels of Gaussian noise to the outputs in the training data. As it turns out, this makes the models indeed more robust against noise. For details, see lines 352ff in the original code of Kamienny et al. at https://github.com/facebookresearch/symbolicregression/blob/main/symbolicregression/envs/environment.py
>
> - **Limited comparison with recent SR models like uDSR, TPSR and PySR**
>
> We have added a comparison to TPSR and PySR to the supplement. TPSR is a trade-off between an E2E transformer and a search-based approach. It loses the advantage of fast inference, while not reaching SOTA recovery. A comparison to uDSR is unfortunately not possible in the limited time frame for the rebuttal, because inference for uDSR is in the range of hours per problem instance.
>
> **It's not clear what are the main features in new data generation setting that lead to this recovery boost? Ablation analysis is needed on the data generation components.**
>
> Both components of our data generation approach, that is, the enumeration of *minimal RMPs* and the *standardization of both inputs and outputs*, are essential for achieving significant recovery rates. Without the first component, that is, minimal RMPs, we would miss too many equivalence classes of low complexity functions. By the standardization, we avoid a shortcoming in the Kamienny approach, which requires rescaling the outputs for prediction, which includes fitting many constants. As we have discussed before, we could have also used minimal expression trees in prefix notation instead of minimal RPMs for the training. It is the minimality that is important.
>
> **Minor comment: What beam size were used for E2E and E2E-RMP results?**
> For E2E as well as for E2E-RMP a beam size of ten was used.

---

> ### Comment · Reviewer_W4ae · 2024-11-25
> **Response to Authors**
>
> Thank you for your response. However, several critical issues still remain unaddressed.
>
>
> > (1) in-domain E2E-RMP has a recovery rate that is ~7% better than [1], which we explain by the systematic enumeration of equivalence classes of functions with succinct representatives (minimal RMPS), and (2) out-of-domain E2E-RMP degrades significantly more gracefully than [1], which we explain by the in- and output standardization. Note also that, in contrast to us, [1] does contain all Feyman problems in their training set.
> >
> The new comparative results with (Biggio et al., 2021) show that the improvements are marginal at best. This makes me wonder that maybe the main benefit of the proposed method E2E-RMP is coming from the skeleton-based training (which was already introduced by Biggio et al) rather than other important data design components (such as RMP). This significantly diminishes the claimed technical contributions of this work, especially considering that this crucial baseline was not even cited in the original submission.
>
>
> > We have added a comparison to TPSR and PySR to the supplement.
> >
> New comparison results with state-of-the-art methods in Table 3 Appendix shows that in terms of recovery rate in SRBench problems, the proposed method E2E-RMP is having recovery rate 34.02 which is only slightly better than TPSR with recovery 33.89 and worse than PySR with recovery rate 44.73. Authors might raise that PySR is an evolutionary search method and is not based on Transformers pre-training. What about TPSR? TPSR is only searching on top of E2E Transformer model and it's already obtaining very similar results to the proposed method in this paper in terms of recovery and even better performance in terms of R2 (0.99 for TPSR vs 0.72 for E2E-RMP as reported by authors in the Table 3).
>
> Also, due to the similarity of TPSR results to the proposed method in terms of recovery, I think TPSR also needs to be added to Figure 9 for detailed recovery comparisons.
>
>
> > For training the transformer, it is not important if the expressions are represented by RMPs or by expression trees in prefix notation, as long as they are minimal.
> >
> This response raises a fundamental question about the paper's contribution. If RMPs are not providing significant advantages over expression trees, why introduce this additional complexity? The logical conclusion would be to find minimal expressions using RMPs and then use their corresponding expression trees for training, as in E2E. This undermines one of the paper's main claimed contributions.
>
>
> > Both components of our data generation approach, that is, the enumeration of minimal RMPs and the standardization of both inputs and outputs, are essential for achieving significant recovery rates.
> >
> Is there any experimental evidence to support this? I can't find any evidence in the paper regarding the impact of detailed data generation components on performance. Without that, it's hard to understand what component is essential and is causing benefit in the proposed data generation setting, or if the benefit is only coming from a skeleton-based model already established in prior work (as in Biggio et al.)
>
> ---
> Given these unaddressed concerns and the new comparative results revealing limited improvements over SOTA methods, I am lowering my score for this work.

---

> > ### Author Response · Authors · 2024-11-25
> >
> > Thank you for your feedback. We still want to clarify three points:
> >
> > **New comparison results with state-of-the-art methods in Table 3 Appendix shows that in terms of recovery rate in SRBench problems, the proposed method E2E-RMP is having recovery rate 34.02 which is only slightly better than TPSR with recovery 33.89 and worse than PySR with recovery rate 44.73. Authors might raise that PySR is an evolutionary search method and is not based on Transformers pre-training. What about TPSR? TPSR is only searching on top of E2E Transformer model and it's already obtaining very similar results to the proposed method in this paper in terms of recovery and even better performance in terms of R2 (0.99 for TPSR vs 0.72 for E2E-RMP as reported by authors in the Table 3). Also, due to the similarity of TPSR results to the proposed method in terms of recovery, I think TPSR also needs to be added to Figure 9 for detailed recovery comparisons.**
> >
> > In contrast to other transformer-based methods which have inference times in the range of milliseconds, TPSR has inference times in the same range as search-based methods (10^3 seconds). These search-based methods achieve significantly higher recovery than TPSR. That is, TPSR is neither fast at inference time nor efficient in terms of recovery.
> >
> > **This response raises a fundamental question about the paper's contribution. If RMPs are not providing significant advantages over expression trees, why introduce this additional complexity? The logical conclusion would be to find minimal expressions using RMPs and then use their corresponding expression trees for training, as in E2E. This undermines one of the paper's main claimed contributions.**
> >
> > The key point is that we need *minimal* RMPs or *minimal* expression trees because otherwise we will miss equivalence classes of functions that have succinct representatives. RMPs are more succinct than expression trees because they incorporate common subexpression elimination. Enumerating minimal expression trees entails common subexpression elimination, rendering it much more inefficient. That is, our approach is much more efficient for enumerating succinct representations because it avoids the extremely costly common subexpression elimination.
> >
> > **Is there any experimental evidence to support this? I can't find any evidence in the paper regarding the impact of detailed data generation components on performance. Without that, it's hard to understand what component is essential and is causing benefit in the proposed data generation setting, or if the benefit is only coming from a skeleton-based model already established in prior work (as in Biggio et al.)**
> >
> > Biggio et al. randomly sample expression trees instead of systematically enumerating succinct representations. Therefore Biggio et al. miss equivalence classes with succinct representations. As can be seen in Table 3 in the supplement, this results in a significantly lower recovery rate.

---

### Official Review · Reviewer_195R · 2024-11-03

**Soundness:** 3
**Presentation:** 3
**Contribution:** 3
**Rating:** 6
**Confidence:** 3

**Summary:**

The paper proposes a a structured way to  select the training data and construct a synthetic dataset from first principles, improving the interpretability and efficiency of symbolic regression using transformer models.

**Strengths:**

1. The article is clearly written and its structure is well-organized

2. The experimental validation in the article is sufficient.

3. The method proposed in the article is practical and demonstrates good effectiveness and innovation.
    What’s most valuable is that the method integrates well with the hardware.

**Weaknesses:**

The paper highlights the transformer model’s tendency to memorize training data, as 51% of the recovered formulas are from the training set. This method is somewhat too direct, which may lead to overfitting of the model.

**Questions:**

How does this method achieve fine-grained control of registers to complete a series of complex computational tasks?

**Details Of Ethics Concerns:**

None.

---

> ### Author Response · Authors · 2024-11-13
> **Rebuttal**
>
> Thank you for your review. Here, we answer your question.
>
> **How does this method achieve fine-grained control of registers to complete a series of complex computational tasks?**
> The model generates the register machine programs sequentially, token by token. The registers are named $a_i$, while the inputs are named $x_i$ and the constants are named $c_i$. In order to control the registers in a series/sequence of computations, the model can reuse existing results from previous registers by referring to the register name.

---

### Official Review · Reviewer_ZRDu · 2024-11-04

**Soundness:** 3
**Presentation:** 3
**Contribution:** 3
**Rating:** 6
**Confidence:** 4

**Summary:**

This paper aims to improve the recovery performance of transformer-based symbolic regression methods through more systematic training data generation. The authors represent equations as register machine programs (RMPs) and propose finding minimal RMPs with standardized input/output variables. The approach shows improved recovery rates on simpler equations from benchmarks like Feynman equations while maintaining the fast inference advantage of transformer-based approaches.

**Strengths:**

- Addresses an important limitation in transformer-based symbolic regression methods by focusing on their poor recovery performance
- The proposed minimal RMP generation approach shows promise in reducing expression complexity
- Demonstrates meaningful improvement in recovery rate over E2E methods for simpler Feynman problems while maintaining fast inference times

**Weaknesses:**

- Limited evaluation on complex problems - the approach's effectiveness seems primarily demonstrated on simpler equations
- The standardization approach, while helpful for training, may limit the model's ability to handle complex nonlinear relations with varying constant ranges
- The empirical advantages over E2E approach are not clearly demonstrated through comprehensive metrics (e.g., R² accuracy, black-box problems), especially given that E2E's data generation could potentially be adjusted to achieve similar results

**Questions:**

* Do minimal RMPs have equivalent expression trees? Can you comment on potential performance if the model was trained on expression tree versions of the same minimal RMP datasets?

* Figure 4 presents recovery rate results. Could you provide:
   - R² accuracy performance on SRBench problems (both ground truth and black-box functions)
   - Fitting accuracy (R²) comparison across different dimensions relative to E2E

* In Figure 5, why does recovery performance drop with context length? Shouldn't more observations improve the performance?

* Given that data generation includes RMP enumeration for finding minimal RMPs, how time-consuming is the data generation process?

* You report that in 10K equations, E2E contains around 6.4K equivalence groups, and E2E prioritizes fast data generation. Could E2E generate the same number of equivalent groups given equal time?

* One drawback of E2E data noted in the paper is its tendency toward complex forms, especially compared to low-dimensional Feynman equations. However, E2E's data generation can be controlled through hyperparameters (minimum/maximum number of unary and binary operators). Have you explored if reducing equation length in E2E could lead to simpler equations and better recovery, even if it comes at the cost of lower accuracy?

* Would it be possible for the authors to share the code for data generation and model weights?

* Could you explain:
   - Why mRMPs per dimension decreases from D=3 to D=5 in Appendix Figure 3?
   - Is the constant range [-10, 10] reasonable for practical problems, given E2E uses [-100, 100]? Can the model generalize to out-of-range constants?
   - Could you provide examples of the 17% of recovered formulas that result from generalization (Lines 409-412)?
   - What does "implicitly" vs "explicitly" mean in Lines 077-079?
   - In Line 200-203, why is c₃ ∈ ℝ? Shouldn't it be c₃ ∈ ℝᴰ?

[Note: There appears to be a typo in Line 285 where "RPM" should be "RMP"]

---

> ### Author Response · Authors · 2024-11-14
> **Rebuttal**
>
> Thank you for your review. Before we address your questions, we would like to comment on the weaknesses mentioned.
>
> **Eval. on complex problems**
>
> The recovery performance of SOTA symbolic regressors degrades with growing function complexity, which is typically measured by the size of a minimal expression tree. A lower bound on the complexity is two times the number of input variables. The E2E symbolic regressor is no exception to this rule. We have added a plot to the supplement that shows this phenomenon for various SOTA symbolic regressors.
>
> **Limitations of the standardization approach**
>
> The standardization practically affects only the shift and the scale of the output. In practice, those effects can be dealt with by optionally fitting a constant to the generated expression.
>
> **Unclear advantages over E2E**
>
> The performance of the E2E approach depends on both the architecture and the training data generation. We keep the architecture of the original E2E approach, and only differ in the data generation method. Our novel data generation method empirically shows a drastically improved recovery.
>
> ### **Questions and answers**
>
> **Do minimal RMPs have equivalent expr. trees?**
>
> Yes, for each RMP there exists an equivalent expression tree. In RMPs, however, subexpressions can be reused, which makes for more compact representations. Therefore, we can express the same expression more succinctly. Once the minimal RMPs have been identified, it does not make a difference, whether we train on the expression trees or RMPs.
>
> **For Figure 4, could you provide:**
> - **R² accuracy performance SRBench**
>
> The R2 accuracy performance on SRBench ground truth problems is already provided in the supplement in Figure 6. However, in the context of symbolic regression, we consider the accuracy-complexity trade-off as shown in Figure 7 of the supplement more important than a pure accuracy measure such as R2, because low complexity leads to better interpretability.
>
> - **Fitting acc. (R²) relative to E2E**
>
> The accuracy-complexity tradeoff for E2E and E2E-RMP is shown in Figure 7 (left) of the supplement. E2E-RMP is only slightly worse in terms of accuracy (R2) compared to E2E, but the generated expressions are far less complex and more interpretable.
>
> **In Figure 5, why does recovery performance drop with context length?**
>
> We see a recovery decrease with increasing input length, larger models decrease much less (Table 5 supplement). We conjecture that increasing the input length while keeping the parameter size fixed increases the required computational complexity for the model to make an accurate prediction.
>
> **How time-consuming is the data generation process?**
>
> Enumerating and checking RMPs with a maximum length of five lines and a maximum number of five input variables took approx. one hour on an Intel(R) Xeon(R) Gold 6226R CPU @ 2.90GHz.
>
> **Could E2E generate the same number of equivalent groups given equal time?**
>
> Even if the standard E2E sampling process would cover the same number of equivalence classes given the same time, the problem still would be that these classes are scattered over many complexity levels and therefore miss many low complexity equivalence classes, representing interpretable functions.
>
> **Have you explored reducing equation length?**
>
> Systematically enumerating succinct expressions, which in a sense means small equation length, is exactly what we do in our method. The other important contribution is the standardization of input and output values. Note that in the noise-free setting, recovery implies perfect accuracy. Therefore, one cannot improve recovery at the cost of reduced accuracy.
>
> **Would it be possible to share the code?**
>
> We plan to make the code publicly available on GitHub.
>
> **Could you explain**
> - **Why mRMPs per dimension decreases from D=3 to D=5?**
>
> This is implied by the combinatorics of the problem.  When the number of dimensions increases, then more binary operators (addition or multiplication) are needed to generate valid RMPs. Every additional input requires one more operator to be a binary operator, which reduces the number RMP lines, where the operator can be chosen freely (unary or binary).
>
> - **Is the constant range reasonable for practical problems?**
>
> The model does not predict constants but placeholders. Therefore, it does not need to generalize to out of range constants. Placeholders are replaced by actual constants by a fitting procedure. During training, the constant range can be set by the user to any interval $[a,b]$.
>
> - **Examples of the recovered formulas**
>
> We added the recovered formulas that result from generalization to the supplement.
>
> - **"implicitly" vs "explicitly"?**
>
> Explicitly means enumerating all small expressions, whereas implicitly means steering a search towards small expressions by a search policy.
>
> - **In Line 200-203, why is $c_3\in\mathbb{R}$?**
>
> Here, we scale the vector $x\in\mathbb{R}^D$ by the scalar $c_3\in\mathbb{R}$.

---

> > ### Comment · Reviewer_ZRDu · 2024-11-21
> > **Response to Authors**
> >
> > Thank you for your response to my comments and questions. However, some of my concerns are not addressed.
> >
> > > Therefore, we can express the same expression more succinctly. Once the minimal RMPs have been identified, it does not make a difference, whether we train on the expression trees or RMPs.
> > >
> >
> > * Can you clarify if this statement of "It does not make a difference" comes from the results of experiments, or is this the intuition of authors to the problem? Also, I do not agree with "more succinct" representation using RMPs. As Reviewer XjUE has mentioned, RMPs can indeed use more tokens in many cases compared to the expression trees.
> >
> > * From original review: Could you provide R² accuracy performance (+ complexity) on SRBench **black-box** problems?
> >
> > * Reducing equation length in E2E: My concern has been that by reducing equation length in E2E training data by simple hyper parameter tuning, one might see the similar results of improving recovery by generating equation of lower complexity. It would be great to see experimental results on training E2E with lower equation lengths in training data.
> >
> > > The model does not predict constants but placeholders. Therefore, it does not need to generalize to out of range constants.
> > >
> >
> > * My concern regarding the constant range is that when the model is trained on smaller range of constants, the distribution is much more limited, which can happen to be close to the distribution of target experimental benchmarks such as simple Feynman equations.
> >
> > * Dimensionality of c3: The input features in a symbolic regression task might have different ranges (consider physical variables), and they need different scaling factors in each dimension. So, scaling all features using a scalar is not the best approach.
> >
> > * Performance drop with context length: The authors' response does not align with observations from previous works (e.g., see Fig 4.D in E2E (Kamienny et al.) paper), as well as common ML models, where increasing data observations (evidence) improve the model performance.
> >
> > * In Figure 9 in supplements (comparison of recovery vs. complexity), where does E2E-RMP stands in this plot?

---

> ### Author Response · Authors · 2024-11-21
>
> Thank you for your feedback on our rebuttal.
>
> **Can you clarify if this statement of "It does not make a difference" comes from the results of experiments, or is this the intuition of authors to the problem? Also, I do not agree with "more succinct" representation using RMPs. As Reviewer XjUE has mentioned, RMPs can indeed use more tokens in many cases compared to the expression trees.**
>
> Instead of tokenizing RMPs and feeding them into the transformer, we can also expand RMPs into expression trees, tokenize them, and feed the tokenization into a transformer. However, RMPs are more succinct than expression trees, because they incorporate common subexpression elimination, see also [1,2]. Here, succinctness is measured as the number of (unary and binary) operators, but not as the number of tokens used in the tokenization scheme. For tokenizing RMPs, we use a tokenization scheme that aims for clarity and ease of use, because this promotes efficient encoding and decoding of RMPs.
> However, the more important advantage of RMPs is that we have a systematic algorithm for enumerating minimal RMPs. Note that enumerating minimal expression trees entails common subexpression elimination, rendering it much more inefficient.
>
> [1] P. J. Downey, R. Sethi, and R. E. Tarjan. Variations on the common subexpression problem. Journal of the Association for Computing Machinery, 27(4):758–771, 1980.
>
> [2] P. Flajolet, P. Sipala, and J.-M. Steyaert. Analytic variations on the common subexpression problem. In Proceedings of ICALP 1990, volume 443 of Lecture Notes in Computer Science, pages 220–234. Springer, 1990.
>
> **From original review: Could you provide R² accuracy performance (+ complexity) on SRBench black-box problems?**
>
> Here, we consider recovery as a quality metric for the interpretability of symbolic regressors. Therefore, our goal is to measure and boost the recovery performance of (transformer-based) symbolic regressors. Since, by definition, black box problems do not come with ground truth formulas, they cannot be used for measuring recovery performance. However, we will add data on the accuracy-complexity tradeoff to the supplement.
>
> **Reducing equation length in E2E: My concern has been that by reducing equation length in E2E training data by simple hyper parameter tuning, one might see the similar results of improving recovery by generating equation of lower complexity. It would be great to see experimental results on training E2E with lower equation lengths in training data.**
>
> Indeed, low equation length means small RMP. However, by randomly sampling equations with small equation length has the problem that one can still miss equivalence classes of functions with succinct representations. Therefore, it is important not to sample but to systematically enumerate equations with small equation length. This is exactly what we are doing by enumerating minimal RMPs.
>
> **My concern regarding the constant range is that when the model is trained on smaller range of constants, the distribution is much more limited, which can happen to be close to the distribution of target experimental benchmarks ...**
>
> This is a common problem of all transformer-based approaches that all have to specify the input domain and the constant range when generating the training data. As we have shown, the problem can be mitigated by the input-output standardization that we proposed in the paper.
>
> **Dimensionality of c3: The input features in a symbolic regression task might have different ranges (consider physical variables), and they need different scaling factors in each dimension. So, scaling all features using a scalar is not the best approach.**
>
> You might be confusing two things. One is the standardization of data sets that, indeed, might need different scaling factors per dimension. The other is our notion of the *affine equivalence* of two functions, where we consider two scalar functions equivalent, when they differ by a scaling factor (and translations).
>
> **Performance drop with context length: The authors' response does not align with observations from previous works (e.g., see Fig 4.D in E2E (Kamienny et al.) paper) ...**
>
> Here, the input length (which, as pointed out by Reviewer XjUE is notationally better than context length) is the number of input-output pairs per function as seen by the transformer. Indeed, for a fixed input domain the recovery performance for the transformer improves, when the test data are sampled from the same input domain. But, crucially, it does not improve, when the test data are sampled out of domain. As can be seen in Table 3 of the supplement, even, within domain, our approach performs much better than standard E2E approach, because of the input- and output standardization.
>
> **In Figure 9 in supplements (comparison of recovery vs. complexity), where does E2E-RMP stands in this plot?**
>
> Thank you for pointing this out.  We will update the supplement with E2E-RMP added to the plot.

---

### Note · Authors · 2025-01-18

**Comment:**

We thank all the Reviewers and Area Chairs for their time and effort.

Based on the reviews, we have significantly revised our paper and would like to withdraw the submitted version.

**Withdrawal Confirmation:**

I have read and agree with the venue's withdrawal policy on behalf of myself and my co-authors.